# A Comprehensive Review of Mammalian Pigmentation: Paving the Way for Innovative Hair Colour-Changing Cosmetics

**DOI:** 10.3390/biology12020290

**Published:** 2023-02-11

**Authors:** Bruno Fernandes, Artur Cavaco-Paulo, Teresa Matamá

**Affiliations:** 1CEB—Centre of Biological Engineering, University of Minho, Campus of Gualtar, 4710-057 Braga, Portugal; 2LABBELS—Associate Laboratory, 4710-057 Braga, Portugal

**Keywords:** hair follicle, melanocytes, melanosomes, melanogenesis, melanins, hair pigmentation, hair colour, cosmetics

## Abstract

**Simple Summary:**

The frequency of use of permanent hair dyes to change natural colour are associated with fibre damage and with an increased risk of serious health problems. The future of hair dyeing could be the topical modulation of the pigment production that occurs in special cells, called melanocytes, localized in the hair roots. The success of such approaches is dependent on expanding and deepening our knowledge on hair pigmentation biology. In this context, the paper aims to critically review the vast bibliography on mammalian pigmentation having in view the topical modulation of hair follicle biology to produce a desired change in the hair fibre colour.

**Abstract:**

The natural colour of hair shafts is formed at the bulb of hair follicles, and it is coupled to the hair growth cycle. Three critical processes must happen for efficient pigmentation: (1) melanosome biogenesis in neural crest-derived melanocytes, (2) the biochemical synthesis of melanins (melanogenesis) inside melanosomes, and (3) the transfer of melanin granules to surrounding pre-cortical keratinocytes for their incorporation into nascent hair fibres. All these steps are under complex genetic control. The array of natural hair colour shades are ascribed to polymorphisms in several pigmentary genes. A myriad of factors acting via autocrine, paracrine, and endocrine mechanisms also contributes for hair colour diversity. Given the enormous social and cosmetic importance attributed to hair colour, hair dyeing is today a common practice. Nonetheless, the adverse effects of the long-term usage of such cosmetic procedures demand the development of new methods for colour change. In this context, case reports of hair lightening, darkening and repigmentation as a side-effect of the therapeutic usage of many drugs substantiate the possibility to tune hair colour by interfering with the biology of follicular pigmentary units. By scrutinizing mammalian pigmentation, this review pinpoints key targetable processes for the development of innovative cosmetics that can safely change the hair colour from the inside out.

## 1. Introduction

Throughout human history, people have changed their hair colour as a means to segregate their social status. Nowadays, regardless of economical and education backgrounds, millions of individuals worldwide commonly dye their hair to enhance youth and beauty and to follow fashion trends. Because hair colouration has become very popular, hair colouring products now represent one of the most rapidly growing beauty and personal care markets [1,2]. Regarding the year 2019, the global hair colour market was valued at approximately 22.2 billion USD, and it is expected to generate a revenue of around 37.4 billion USD by 2026 (Zion Market Research, NY, USA).

Hair dyeing systems can be divided into oxidative and non-oxidative. Additionally, according to the colour durability, they are classified into temporary, semi-permanent, and permanent [1]. Temporary non-oxidative dyes are only deposited on the hair surface and start leaving the fibre after the first wash. The semi-permanent non-oxidative dyes also interact predominantly with the cuticle (the most external part of the hair strand), with a small amount of dye penetrating the hair cortex; this kind of colouration resists a few washes. Temporary and semi-permanent products do not require chemical reactions to impart colour as they rely on van der Waals forces for adhesion of direct dyes to the hair fibre. Regarding permanent oxidative products, hair colouration happens upon reactions between colourless precursors (developer and coupler) in the presence of an oxidizing agent (e.g., H_2_O_2_) and under alkaline conditions (e.g., ammonia). The combination of oxidizing and alkaline agents causes swelling of the hair cuticle, facilitating the diffusion of the small precursors into the fibre and the bleaching of the natural melanin pigments. Then, the colourless precursors undergo oxidation to form large, coloured molecules that become trapped inside the fibre cortex [3]. The permanent oxidative hair colour products provide greater efficacy of dyeing and are the most used, representing about 80% of the hair-colouring market [2].

Due to the destructive nature of permanent dye products, the hair fibre structure suffers cumulative damages that lead to split ends and dry and dull hair [4]. More important, those products have been identified as the source of various adverse health effects, such as chemical and allergic reactions that many times result in acute or mild dermatitis with consequent hair loss [2,5,6,7,8]. Even more alarming is the fact that some studies have raised the possibility that the long-term usage of permanent dyes can cause serious and systemic side-effects, such as the increased risk of developing certain cancers [9,10,11,12,13]. Despite the inconvenience and liability, in the absence of other ways, many people continue to dye the hair for cosmetic purposes. Thus, the development of safer ways for hair colour modification is more than ever a pertinent issue. The success of such innovative ways for hair colour modification will be closely related to the understanding of natural hair pigmentation biology. The aim when writing this review was to compile the dispersed and most essential information, providing the most comprehensive and recent overview on mammalian melanogenesis.

Natural hair colour is one of the most distinctive human phenotypes, with various selective pressures having contributed to the dissemination of an array of natural shades ranging from black, brown, and blond to red. Additionally, grey/white hair is often seen, harbingering the loss of youth [14,15,16,17,18].

Hair and skin pigmentation are a manifestation of the presence of pigments called melanins. Although epidermal melanin had important evolutionary implications (acting as a filter for ultraviolet (UV) light, protecting skin against UV-induced damage), the reasons behind the evolutionary selective pressures for the development of pigmented scalp hair are less clear. Due to the dominant position of fish in the diet of early humans, pigmented scalp hair might have been important to prevent the build-up of toxic metals from fish species. Since toxins/metals can selectively bind to melanin and since hair fibres present fast growth and turnover, pigmented hair shafts allow the body to rapidly get rid of harmful substances, limiting their access to the living tissue of the highly vascularized scalp. Hair analyses corroborate this theory as significant amounts of various heavy metal ions bound to melanin have been reported [19]. Moreover, as our nakedness draws much attention to our facial and scalp hair, the development of pigmented scalp hairs could also have been favoured as a non-verbal mean of social communication and mate selection [15,16,17,20].

The colouration of hair shafts results from precise and sequential interactions between different cell populations in the hair follicle, the mini-organ responsible for hair shaft production. First, melanin is synthesized by melanocytes, inside cytoplasmic membrane-bound organelles called melanosomes. Melanin is the product of a complex biochemical pathway called melanogenesis and it can undertake two chemically distinct forms: the black to brown eumelanic form and the reddish-brown to yellow pheomelanic form. Melanin is then transferred to the surrounding keratinocytes that will proliferate and differentiate originating hair shafts. During the differentiation process, the growing hairs become pigmented. The diversity in hair pigmentation relies mainly on the quantity and ratio of the two types of melanin incorporated into the hair shafts, with other physical aspects of hair fibres intervening only as minor or segmental colour modifiers [14,15,16,17,18,20,21,22].

The genetic basis of hair colour variety has been the subject of many studies and led to the identification of several genes involved in the normal variation of human hair colour. The protein products depending on these *loci*, and their polymorphisms exhibit different patterns of transcription, translation, and functional activity during the pigmentation of hairs. Melanogenesis per se presents a wide range of potential targets with different functions: enzymes, structural proteins, transcription regulators, transporters, receptors, and their ligands. In addition, many intrinsic molecular signals, as the ones produced by keratinocytes, have also been uncovered as intervenient in many aspects of hair pigmentation and its diversity [15,22].

## 2. Development and Cycling of the Hair Follicle Pigmentary Unit

Melanocytes that constitute the follicular and epidermal pigmentary units are derived from the pluripotent cells of the neural crest, a transient component of the ectoderm located between the neural tube and the epidermis (for a more detailed review please consult [23]). Besides melanocytes, the neural crest gives rise to several other types of cells, including neurons and glia cells of the peripheral nervous system, as well as bone and cartilage cells of the head skeleton. The commitment of neural crest cells to the melanocyte lineage and the entire journey of melanoblasts (precursors of melanocytes) during embryogenesis to produce the pigmentary units is regulated by several factors [20,22,24,25,26]. Microphthalmia-associated transcription factor (MITF) appears to be the master regulator of melanocyte identity at the neural crest. MITF regulates the expression of genes that confer melanocyte characteristics to the neural crest cells, such as tyrosinase (*TYR*), tyrosinase related protein 1 (*TYRP1*), and dopachrome tautomerase (*DCT*). In turn, the transcription of *MITF* is regulated by several other factors [27,28,29,30,31,32,33,34,35,36,37,38].

Once melanoblasts begin to express specific markers, they migrate dorsoventrally through the developing embryo and start to proliferate [26]. The migration of melanoblasts is directed by chemotactic signals and patterns of cell surface molecules and receptors expressed by the mobilized melanoblasts or present within the extracellular matrix through which they move, such as the endothelin receptor type B (EDNRB) and its ligand endothelin 3 (ET-3), as well as KIT proto-oncogene receptor tyrosine kinase (KIT) and its ligand stem cell factor (SCF, also known as KITLG), among other factors [26,28,31,39,40,41,42,43,44]. By the seventh week of gestational age, melanoblasts are already in the human epidermis, remaining there until hair morphogenesis begins, two weeks later.

When hair morphogenesis begins, some melanoblasts leave the epidermis and become distributed in the developing hair follicles. Melanoblasts expressing KIT migrate into the SCF-supplying hair follicle epithelium and, once “SCF-positive”, differentiated melanocytes (meaning melanogenically active) target the hair bulb, the only site of melanin production for the pigmentation of hair shafts [45,46]. In the hair bulb, highly melanogenic melanocytes are located above and around the dermal papilla (Figure 1), establishing a functional unit with neighbouring pre-cortical keratinocytes that receive melanin granules and incorporate them into the forming hair shafts [17]. A minor sub-population of poorly differentiated melanocytes (lacking significant pigmentation) is also found in the most proximal and peripheral regions of the hair bulb; however, due to their spatial distribution, they do not transfer melanin to the pre-cortical keratinocytes [47]. Melanogenically active melanocytes are also present at the basal layers of the infundibulum and sebaceous glands (Figure 1). Although the function of these melanocytes is not clear, given their location at the interface with the external environment and the known antimicrobial potential of melanin and its intermediates, they may have an important role in innate immunity and defence. Melanoblasts that do not express KIT invade the outer root sheath and bulge regions. These undifferentiated, amelanotic melanoblasts function as a pool of melanocyte stem cells (MelSC) that are recruited for regeneration of the cycling portion of the pigmentary unit [17,28,46,47,48,49].

Although they are derived from the same precursors, the follicular and epidermal pigmentary units are quite different [17,18,22,50]. In the hair follicle, there is approximately one melanocyte for every five keratinocytes, while the proportion in the epidermis is one to thirty-six on average. Moreover, active melanocytes in the hair bulb are larger, are more dendritic, produce larger melanosomes, have longer dendrites, and have a more extensive Golgi apparatus and rough endoplasmic reticulum (ER) compared to active melanocytes of the epidermis [17,18,22,50]. Melanin produced in the epidermis is degraded almost completely in the differentiating keratinocyte layers, but melanin granules transferred into follicular cortical keratinocytes are minimally digested providing similar pigmentation to the proximal and distal ends of hair shafts [17,18,22,50,51]. This difference in the melanin degradation pattern is usually attributed to the significantly larger size of follicular melanosomes, which influences their uptake by keratinocytes and susceptibility to enzymatic degradation. Both pigmentary units exhibit quite different antigenic profiles, and the immunological differences between them reflect the fact that the epidermal unit is immunocompetent while the follicular unit resides in the immune-privileged hair bulb [52]. Despite all those, the most striking difference is the fact that hair follicle melanogenesis is cyclic and tightly coupled to the hair growth cycle while epidermal melanogenesis is continuous (although easily stimulated after exposure to UV radiation) [17,18,20,22,51,52].

The hair follicle is one of the most complex mini organs of the human body (Figure 1). It consists of an upper permanent region and an inferior region (containing the hair bulb) that undergoes dramatic cyclic transformation, renewing itself during our lifetime [18,22]. The hair bulb is composed of two main cell types: dermal papilla cells with a mesenchymal embryonic origin and epithelial cells derived from the surface epithelium [53]. The purpose of hair cycling is thought to be related to cleaning the skin surface of debris and parasites, the excretion of deleterious chemicals encapsulated within trichocytes, the regulation of paracrine and endocrine secretion of hormones and growth modulators produced within the hair follicle, and as a safe-guarding system against malignant degeneration by protecting rapidly dividing keratinocytes from oxidative damage by depletion [18,22]. The continuous cycling throughout adult life recapitulates embryonic hair follicle development. Changes in the local milieu, meaning the reciprocal epithelial–mesenchyme interactions governed by several signalling pathways, such as the Wnt/β-catenin, hedgehog, notch, and bone morphogenic protein (BMP) pathways, are thought to drive the hair cycle progression through its different phases: anagen, catagen, and telogen [53]. The activity of hair bulb melanocytes is tightly coupled to the hair growth cycle and undergoes concomitant transformation (Figure 2) [54,55].

Hair shaft synthesis and pigmentation take place during anagen (or growth phase); in the human scalp, 85% to 90% of hairs are in the anagen phase, which lasts on average 3 years. The mature anagen hair follicle contains a fully formed hair bulb. Embraced by the hair bulb lies the follicular papilla of which the volume, number of cells, and secretory activity determine the size of the hair bulb, the duration of anagen, and the length and thickness of hair shafts [18,22,56]. Anagen is divided into stages I to VI, throughout which melanocytes experience several morphological and biochemical changes. At the start of anagen, immature melanocytes located in the permanent part of the hair follicles are stimulated and respond by increasing their volume, entering mitotic activity, and migrating in a coordinate manner, to repopulate the bulb of the new forming hair follicles [17,18,22,55,57]. By anagen II, melanocytes are in a state of proliferation, which becomes quite significant at anagen III and remains until anagen VI (full anagen). In anagen II, tyrosinase activity and melanin also become readily detectable, increasingly rapidly through anagen III to V, reaching their highest levels at early anagen VI. During the progression of anagen III into VI, melanocytes develop more expansive Golgi and rough ER (needed for the melanosome biogenesis), increase the size and number of their melanosomes, and become highly dendritic. Moreover, melanocytes take up position in the hair bulb matrix/dermal papilla border and begin to transfer mature melanosomes into keratinocytes. At anagen VI, as proliferation ceases, bulbar melanocytes assume a highly differentiated status and become fully functional with respect to melanin synthesis. By the end of anagen VI, the earliest signs of imminent hair follicle regression are seen: L-Tyr availability declines, the expression and activity of tyrosinase and other melanogenesis-related proteins decrease rapidly, melanogenesis is attenuated, and melanocytes retract their dendrites [17,18,20,22,54,55,57,58,59].

During the transition from anagen to catagen, tyrosinase is no longer detectable in the hair bulb and the production of melanin is terminated [18,22,55,57]. As the catagen phase proceeds, an extensive apoptosis-driven involution process occurs, causing the loss of up to 70% of the hair follicle in a relatively brief period (2–4 weeks in the human scalp) [18,22,56,60,61]. The current view suggests that melanocytes are recruited, every new cycle, from the base of the permanent part of the hair follicles to populate the hair bulb. This region constitutes a reservoir of immature, slow-cycling, self-maintaining MelSCs that are fully competent to regenerate the pigmentary system of new hair bulbs at the beginning of each anagen phase [17,18,20,54,57,60,61]. Notch signalling, the Wingless-related integration site (Wnt) signalling pathway, paired box 3 (PAX3), MITF, and its downstream target BCL2 have all been proven to play critical roles in the maintenance of these MelSCs by promoting their survival [62,63,64,65].

Finally, telogen is a phase of relative proliferative quiescence, and in the human scalp, it lasts around 3 months [18,22]. Despite that, it is characterized by intricate and functionally relevant biochemical activity, centred around the maintenance of several stem cell populations. These melanocyte precursors are commonly small, have high nuclear/cytoplasmic ratios, an inactive cytoplasm with very few organelles, and a limited number of dendrites, and do not synthesize melanin or express relevant melanogenesis-related proteins [66]. By late telogen, the preparation of hair follicles for imminent regeneration (anagen development) is also characterized by intense biochemical activity. The active shedding of an old hair fibre (club hair) is a process that occurs as a distinct phase called exogen [67,68].

## 3. Molecular and Cellular Biology of Hair Pigmentation

Several critical steps must proceed in perfectly synchrony to achieve the uniform and accurate pigmentation of hair shafts. These steps involve melanosome biogenesis and the biochemical synthesis of melanin in follicular melanocytes, as well as the transfer of melanin granules into surrounding keratinocytes. All these steps are under complex genetic control, with several genes encoding a wide range of structural proteins, enzymes, ion channels and transporters, receptors, and growth factors, among others (Table 1).

MITF is the regulator of many genes associated with the pigmentary process; additionally, it also controls the expression of many other genes involved in differentiation, proliferation, and apoptosis [81,82,83]. MITF is a basic helix–loop–helix leucine zipper (bHLH-ZP) transcription factor that belongs to the MYC superfamily. At least 10 isoforms of MITF have been described in humans, with MITF-M being the specific isoform of the melanocyte lineage [83]. As with other bHLH transcription regulators, it recognizes the nucleotide consensus sequence CANNTG (E-box), with the association being maximal if this motif is 5′-flanked with T. At the N-terminus (NTAD), MITF exhibits the transcription activation domain, essential for interactions with the transcriptional coactivators p300/CREB-binding protein (CBP) [84]. In turn, the transcription of *MITF* is controlled by a variety of regulators capable of binding to specific DNA sequences in its M promoter region: SRY-related HMG-box (SOX) 9, SOX10, cAMP response element-binding protein (CREB), PAX3, lymphoid enhancer-binding factor 1 (LEF1), zinc finger E-box binding protein 2 (ZEB2), one cut domain 2 (ONECUT2), and MITF itself [76,85,86,87,88,89,90].

### 3.1. Melanosomes Biogenesis

Melanosomes are intracellular membrane-bound organelles, produced through the action of the Golgi and rough endoplasmic reticulum, inside which melanin synthesis takes place [91]. They are lysosome-related organelles, part of the secretory/endocytic pathway and sharing many common features with lysosomes [acidic luminal pH, presence of lysosomal-associated membrane proteins (LAMPs) in their outer membrane, among others] [92,93,94]. Despite the many common features, melanosomes also express an array of exclusive proteins (melanogenesis-related proteins, as well as certain LAMPs), and they present distinctive morphological characteristics (manifestation of an ellipsoid shape and internal structures composed of lamellae or regularly striated filaments). Therefore, it is presumed that the biochemical process of melanosome biogenesis forms a separate lineage from the secretory/endocytic pathway [92,93,94].

The development and maturation of melanosomes proceeds through four stages, characterized by unique ultrastructural morphology (Figure 3). These stages can be grouped into two main steps: unpigmented and pigmented [91,93]. The unpigmented step comprises stages I and II during which the melanosome matrix is formed; at this point, melanosomes are referred to as immature or pre-melanosome compartments. The pigmented step comprises stages III and IV, in which the synthesis of melanin starts upon the formed matrix; at this point, melanosomes are denoted as mature/late compartments. Melanosomes are assembled in the perinuclear region, near the Golgi stacks, receiving all enzymatic and structural proteins required for melanogenesis [95,96].

In Stage I, immature melanosomes are spherical vacuolar domains of early endosomes that harbour intralumenal vesicles (ILVs) formed by invagination of the limiting membrane. The irregular arrays of amyloid fibrils that emanate from the ILVs make them distinct from early endosomes in other cells. Those fibrils are mainly composed of fragments of premelanosome protein (PMEL) [85,91,95,96,97,98,99]. In Stage II, pre-melanosomes are characterized by a fully formed melanosome matrix. PMEL fibrils are organized into arrays of parallel sheets, transforming the spherical Stage I melanosomes into elongated, elliptical organelles. Although no active melanogenesis takes place in Stage II, they already contain melanogenic proteins (TYR, TYRP1, among others). The formation of PMEL fibrils segregates the melanosomal from the endosomal pathways. Although the mechanism regulating the separation is unclear, it seems to involve GPR143 (G protein-coupled receptor 143) and MLANA (protein melan-a) [97,98,99,100,101,102]. In Stage III melanosomes, melanogenesis starts, with the pigment being regularly and uniformly synthesized on the fibrillar matrix. This causes the darkening and thickening of the matrix fibrils. By Stage IV, the melanosomes are saturated with melanin and the internal fibrillar structure becomes completely masked by the pigment; mature melanosomes are ready to be transferred to adjacent keratinocytes [91,93,94,95,100].

Melanosomes that contain mainly eumelanin, often called eumelanosomes, are ellipsoidal and they are generally larger than pheomelanosomes, which contain predominantly pheomelanin [103,104]. In red hair, pheomelanosomes are more spherical and have more irregular shapes than eumelanosomes from black hair [103]. Besides differences in the relative pigment composition and morphology, there are biochemical differences. All melanosomes contain tyrosinase; however, only in eumelanosomes, two additional membrane-bound enzymes, TYRP1 and TYRP2, exist [105]. PMEL expression is also strongly downregulated in cells synthesizing pheomelanins, which might explain the more spherical and irregular shapes of pheomelanosomes when compared to eumelanosomes [104,106].

#### 3.1.1. Major Players in Melanosome Biogenesis

As stated above, the three major players in melanosome development and maturation are PMEL, MLANA, and GPR143; altogether, they form a complex in early stages of melanosome biogenesis assuring their proper composition and structure.

##### Premelanosome Protein (PMEL)

PMEL (also known as PMEL17, gp100, SILV, and ME20) is a type I transmembrane glycoprotein, composed of a short signal peptide, a transmembrane domain, a long luminal N-terminal domain, and a short cytoplasmic C-terminal domain. It is the major functional component of melanosomes and the only melanosomal protein necessary for the formation of very stable, β-sheet-rich oligomeric structures that act as scaffolds upon which melanin is polymerized, condensed, and stored [98]. PMEL fibrils take part in the maintenance of melanosome integrity by preventing highly reactive (cytotoxic) melanin intermediates from freely diffusing within the organelle [102,107]. The sequestering of melanin intermediates has also been shown to accelerate melanin synthesis, and its condensation in PMEL fibrils may even optimize the transfer from melanocytes to keratinocytes. Despite its functional role, the amyloid nature of such fibrils is challenging for melanocytes since an incorrect formation and organization leads to toxicity. Thus, the processing, trafficking, sorting, and fibrillation of PMEL is highly regulated to avoid aggregation in the wrong compartments [100].

PMEL is synthesized in the ER and firstly modified there via removal of the signal peptide, the addition of N-glycosides, and the formation of disulphide bonds. Then, PMEL is exported to the Golgi complex and further modified via *O*-glycosylation. PMEL also undergoes proteolytic processing in the Golgi apparatus and post-Golgi compartment by a proprotein convertase (PC), forming a large fibrillogenic Mα fragment, which remains linked to the Mβ fragment by a disulphide bond [100,108,109,110,111]. From the trans-Golgi network (TGN), PMEL is targeted to endosomes and then to pre-melanosomes where it is further cleaved by the β secretase β-site APP-cleaving enzyme 2 (BACE2); the adaptor-related protein complex 2 (AP-2) is required for the transport of PMEL from the TGN to endosomes and for melanosomal accumulation [112]. The cleavage releases the Mα fragment associated with a portion of the Mβ fragment (MβN) and a C-terminal fragment (CTF). The latter is rapidly degraded in the lysosomes, while the MβN fragment suffers another proteolytic cleavage, required for proper fibril formation. This reaction is known to be mediated by ADAM metallopeptidase domain 17 (ADAM17), a disintegrin and metalloprotease, from a family of proteases known to be involved in ectodomain shedding and for playing a role in cellular processes, such as adhesion and migration [113]; besides ADAM17, other proteins are also thought to be involved in the processing of MβN fragments [114]. Finally, fully processed PMEL fibrillogenic fragments are sorted to the ILVs that, allowing the loading and concentration of such fragments at their surface, provide a favourable environment for the nucleation of fibrils [96,100]. Unlike most proteins, the sorting of PMEL to ILVs does not require ubiquitylation or the activity of the endosomal sorting complexes required for transport (ESCRT) machinery, being regulated by apolipoprotein E (APOE) and tetraspanin CD63 [115,116]. Along with the maturation of PMEL fibrils into sheets, the ILVs disappear, probably by fusing with the membrane of melanosomes or through degradation mediated by lipases [96,100,115,116].

##### Melan-a Protein (MLANA)

MLANA (also known as melanoma antigen recognized by T cells 1 or MART-1) is an integral membrane protein. Besides melanosomes, it can be found in late endosomes and lysosomes, among others. MLANA forms a complex with PMEL, affecting its expression, trafficking, stability, and processing. MLANA also interacts biochemically with GPR143, providing stability to this receptor [96,101,117].

##### G protein-Coupled Receptor 143 (GPR143)

GPR143 (also known as oculocutaneous albinism 1 protein or OA1) is a pigment-cell specific transmembrane G protein-coupled receptor found in the membrane of melanosomes and other organelles, such as late endosomes and lysosomes [118,119]. This receptor acts as a biosensor of melanosome maturation and regulates their size and number by triggering the activation of a transcription cascade, involving MITF, which culminates in sustained PMEL expression [120]. It has also been proposed that GPR143 activity delays the delivery of PMEL-containing endosomes to the lysosomes in order to allow time for the commitment to melanosome biogenesis [119]. Like many membrane G protein-coupled receptors, GPR143 might also be coupled to ion channels and modulate ion gradients and membrane potential in melanosomes, important in fusion and fission events that regulate protein trafficking and organelle biogenesis [121].

#### 3.1.2. Protein Trafficking to Melanosomes

The maturation of melanosomes requires the delivery of several melanogenic proteins. Most of them are integral membrane glycoproteins, synthesized by ribosomes associated with the ER, and become integrated in its membrane across which they translocate. In the ER, such proteins are N-glycosylated, folded, and assembled by enzymes and chaperones. Melanosomal proteins that are properly folded in the ER transit to the Golgi complex for functional modifications and from there to the endosomes; failure to be properly processed results in the retro translocation of these melanosomal proteins in the ER and degradation by the proteasome [92,96]. The molecular intervenient of the transition of melanogenic proteins from the Golgi to endosomes is mainly unknown, but the interaction between GAIP-interacting protein COOH-terminus (GIPC) and adaptor protein containing pleckstrin homology, phosphotyrosine binding domain, and leucine zipper motif (APPL) seems to be important, at least in the trafficking of TYRP1 [122,123].

Within endosomes, melanogenic proteins are usually loaded into vesicular or tubular carriers for delivery into melanosomes through a process involving the fusion of their limiting membranes. The sorting and transport of melanogenic proteins involve multisubunit protein complexes (Figure 3). Some of those protein complexes are the adaptor-related protein complexes, namely AP-1 and AP-3 [96]. The AP-1 complex is involved in the trafficking of TYR and TYRP1. The function of AP-1 requires the recruitment of microtubule motor protein kinesin family member 13A (KIF13A) to position endosomes near maturing melanosomes and facilitates the formation of transient connections between them for cargo transfer; two other molecular motor proteins, myosin VI (MYO6) and MYO1B, were also shown to be part of the machinery that regulates the delivery of melanogenic enzymes to maturing melanosomes [96,124,125,126,127]. The AP-3 complex is required for the efficient delivery of TYR and oculocutaneous albinism 2 protein (OCA2), a protein involved in the regulation of melanosomal pH and a major determinant of mammalian pigmentation; OCA2 itself has also been pointed out as being implicated in the trafficking and processing of tyrosinase and related proteins [96,127,128,129,130,131,132,133,134]. Other protein complexes mediating the transport of melanogenic proteins are the lysosomal organelle complexes (BLOC). BLOC-1 allows the delivery TYRP1 and ATP7A to melanosomes; ATP7A is a copper-transporting ATPase that supplies Cu^2+^ to the Cu^2+^-dependent enzyme TYR, thus sustaining melanin synthesis. The trafficking of TYRP1 by this pathway was shown to be dependent on the ESCRT-I complex. Moreover, phosphatidylinositol-4-phosphate and the type II phosphatidylinositol-4-kinases (PI4KIIα and PI4KIIβ) support BLOC-1-dependent tubule formation for protein cargo delivery to stage III melanosomes. The depletion of either PI4KIIα or PI4KIIβ with shRNAs in melanocytes reduced melanin content and misrouted BLOC-1-dependent cargoes to late endosomes/lysosomes. PI4KIIα and PI4KIIβ function sequentially and non-redundantly downstream of BLOC-1 during tubule elongation toward melanosomes by generating local pools of phosphatidylinositol-4-phosphate [135]. BLOC-2 is implicated in the delivery of TYR and TYRP1 to the melanosome [96,136,137,138].

Soluble N-ethylmaleimide-sensitive-factor attachment protein receptors (SNAREs) are also reported to be involved in the delivery of proteins into the melanosomes due to their role in the membranes fusion process. The interaction of melanosomal SNAREs syntaxin 3 and synaptosome-associated protein 23 (SNAP23) with vesicle-associated membrane protein 7 (VAMP7) on TYRP1-containing vesicles regulates its trafficking to melanosomes. Additionally, the interaction of melanosomal VAMP7 with endosomal syntaxin 13 or VPS9-domain ankyrin repeated protein (VARP) is also accountable for the proper delivery of TYR and TYRP1 to melanosomes [96,139,140].

The members of the Rab family of small GTPases are master regulators of intracellular membrane traffic in all eukaryotes, and since most melanosomal cargos are transmembrane proteins, their sorting and transport during melanosome biogenesis is regulated by RAB proteins [141]. RABs have two main functions in melanosome biogenesis: (1) sorting and transport of cargo proteins, such as melanogenic enzymes, to immature melanosomes; (2) movement of melanosomes themselves through interactions with motor proteins [141]. The RAB32/38–VARP complex transports endosomal vesicles that contain either melanogenic enzyme (TYR or TYRP1) and VAMP7, which subsequently promotes fusion between the vesicles and immature melanosomes [141,142,143]. In addition to their involvement in the anterograde transport of melanogenic enzymes, Rab32/38 are involved in VAMP7 recycling from melanosomes through anterograde transport [141,144]. Other RABs are involved in the trafficking of melanogenic proteins. RAB7 has a role in the differential sorting of Tyrp1, tyrosinase, and gp100 from the Golgi to early-stage melanosomes [145,146,147,148]. RAB9 probably participates in the recycling of melanogenic enzymes from stage III melanosomes [141]. RAB11B regulates the direct fusion of the melanosome membrane with the melanocyte plasma membrane [141]. For a more complete perspective on RAB importance and involvement in the trafficking of melanogenic proteins, as well as in other aspects of melanosomes biogenesis, readers should consult [96,141,142,143,145,146,147,148,149,150,151].

### 3.2. Biochemical Pathway of Melanogenesis

Melanogenesis in vivo produces mixtures of two chemically distinct types of melanins through a process often referred to as mixed melanogenesis. Eumelanin is a black-to-brown highly polymerized pigment, while pheomelanin is reddish-brown to yellow and less polymerized. The synthetic pathways for eumelanin and pheomelanin branch from different reactions involving dopaquinone [152]. Pheomelanin was reported to be synthesized first, with eumelanin being subsequently deposited on the preformed pheomelanin, yielding melanin granules. The mixed melanogenesis can be considered a three-step process. Initially cysteinyldopa (CD) formation will occur, as long as the concentration of cysteine is above a certain threshold amount. Second, the oxidation of CD to pheomelanin occurs for concentrations of CD greater than another threshold value. Finally, eumelanin is generated, occurring after most of the CD and cysteine levels are depleted [152]. In hair follicles, the synthesis of pheomelanin can be completely switched-off during eumelanogenesis, but pheomelanogenesis never occurs with the total inhibition of eumelanin synthesis. The total amount of melanins produced is proportional to the availability of L-Tyr and the dopaquinone (DQ) formed, which in turn is proportional to the activity of tyrosinase. The ratio of eumelanin to pheomelanin is also determined by the availability of cysteine [14,15,153,154,155].

The initiation of melanogenesis requires L-Tyr, which can be directly transported from the extracellular space or synthesized inside melanocytes through the hydroxylation of L-phenylalanine by PAH (EC 1.14.16.1). The activity of PAH depends on the essential cofactor/electron donor 6-BH_4_, for which melanocytes have full capacity for de novo synthesis and recycling (Figure 4).

The rate-limiting enzyme in the de novo synthesis of 6-BH_4_ is GTP-cyclohydrolase I (GTP-CH-I, EC 3.5.4.16). GTP-CH-I is stimulated by GFRP, and in turn, GFRP can be stimulated by L-phenylalanine or inhibited by 6-BH_4_ [156,157,158,159]. The recycling of 6-BH_4_ commences with the formation of L-Tyr from L-phenylalanine via PAH. This leads to the generation of 4a-hydroxy-tetrahydrobiopterin (4a-OH-BH_4_), which through a non-enzymatically process, can generate (7R)-L-erythro-5,6,7,8-tetrahydrobiopterin (7-BH_4_). On the other hand, its enzymatic metabolization by 4a-hydroxy-tetrahydrobiopterin dehydratase (4a-OH-BH_4_ DH, EC 4.2.1.96) produces quinonoid dihydropterin (q-BH_2_), which is reduced by the NADH-dependent dihydropteridine reductase (DHPR, EC 1.5.1.34) or reduced glutathione (GSH) to 6-BH_4_. Additionally, thioredoxin reductase (TR, EC 1.8.1.9) controls the redox status of 6-BH_4_/6-biopterin, with implications in the regulation of melanin production, reduced 6-BH_4_ can bind to tyrosinase and inhibit the enzyme, while oxidized 6-biopterin has no effect on it [158,160,161,162].

To ensure that the synthesis of melanin can occur, L-Tyr must be taken into melanosomes. SLC7A5 (solute carrier family 7, member 5) encodes a member of the L-type amino acid transporter family that it is specialized in the transport of histidine, tryptophan, tyrosine, and neutral amino acids. Although the inhibition of this transporter causes the in vitro loss of pigmentation, its presence in melanosomes has not yet been validated and thus, the nature of the tyrosine transporter inside melanosomes remains elusive [163]. In melanosomes, melanogenesis begins with the synthesis of L-3,4-dihydroxyphenylalanine (L-DOPA) from L-Tyr and subsequent oxidation to DQ (Figure 5). The hydroxylation of L-Tyr to L-DOPA is catalysed by the multi-functional enzyme tyrosinase (EC 1.14.18.1), the most important enzyme in the melanogenic pathway. Alternative mechanisms for L-DOPA formation include the reduction of DQ back to L-DOPA or even the direct hydroxylation of L-Tyr by tyrosine hydroxylase isoform I (TH I, EC 1.14.16.2) [22,154,164,165]. From here, mixed melanogenesis proceeds, using the same precursor (DQ) for the synthesis of pheomelanin and eumelanin.

Pheomelanin is spontaneously produced from DQ if cysteine is present in concentrations above 1 μM (Figure 5). The first step is the reductive addition of cysteine to DQ, giving rise to CD isomers 5-S-cysteinyldopa (5-S-CD) and 2-S-cysteinyldopa (2-S-CD) at a ratio of 5 to 1; an alternative route for CD isomer formation is the conjugation of DQ with glutathione, followed by the hydrolysis of the resultant glutathionyldopa by glutamyl transpeptidase. The second step is the redox exchange between CD isomers and DQ to produce CD-quinones (and DOPA), followed by their cyclization through dehydration to form ortho-quinonimine (QI). Then, QI is rearranged with or without decarboxylation to form 1,4-benzothiazine intermediates. The ring closure to yield such intermediates may involve a peroxidase/H_2_O_2_ reaction or TYR-catalysed oxidation; in this context, since it regulates the hydrogen peroxide cellular levels, catalase (EC 1.11.1.6) is an enzyme indirectly affecting the production of pheomelanin. The last step involves the polymerization of benzothiazine intermediates to pheomelanin. The production of pheomelanin is preferred over the production of eumelanin in the presence of CD isomer concentrations above 10 μM [154,166,167,168].

The production of eumelanin begins after most CD isomers and cysteine are depleted (Figure 5). As a highly reactive ortho-quinone intermediate, DQ undergoes spontaneous intramolecular cyclization to give rise to cyclodopa. Then, cyclodopa rapidly undergoes a redox exchange with another DQ molecule to produce one molecule of dopachrome, a relatively stable intermediate, and one molecule of DOPA. In the absence of additional factors, dopachrome undergoes spontaneous decarboxylative rearrangement to form 5,6-dihydroxyindole (DHI) [169]. In the presence of DCT (EC 5.3.3.12), the tautomerization of dopachrome can also form 5,6-dihydroxyindole-2-carboxylic acid (DHICA) [170,171,172]. However, although DCT can be found in the human epidermis and follicular melanocytes of many mammalians, human bulbar melanocytes (at least those of eumelanic phenotypes) do not express this protein and human hairs contain low yields of DHICA [173,174]. Both dihydroxyindoles (DHI and DHICA) are further oxidized and assembled via cross-linking reactions into eumelanin polymers. The oxidative polymerization of DHI is catalysed directly by tyrosinase and indirectly by DQ. The oxidation of DHICA in mice is catalysed by TYRP1, but the human homologue does not act the same way. Although its function is not completely clear, in humans, TYPR1 appears to ensure the appropriate processing and stabilization of tyrosinase, to maintain melanosomal structural integrity and to function as a tyrosine hydroxylase at low concentrations of the substrate. Thus, tyrosinase seems to be responsible for the oxidation and incorporation of DHICA units into human eumelanins, along with peroxidase (EC 1.11.1.7) and PMEL [107,175,176,177,178,179,180,181,182].

#### 3.2.1. Major Players in Melanin Synthesis

As has already been exposed, the synthesis of melanins cannot occur without the activity of TYR and TRP1. In turn, the ratio of pheomelanin to eumelanin is determined by the availability of cysteine, which correlates with the activity of cystine transporters.

##### Tyrosinase (TYR)

Tyrosinase is a type 3 copper-containing enzyme, involved in several steps of melanin synthesis. This membrane glycoprotein can be divided into three domains: a large N-terminal intra-melanosomal domain (that possesses a catalytic subdomain), a single transmembrane α-helix domain (anchoring tyrosinase in melanosome membranes), and a small C-terminal cytoplasmic tail. These three regions are conserved among TYR and TYR-related proteins, and they follow quite similar processing and trafficking pathways. The maturation of tyrosinase includes signal sequence cleavage, heavy glycosylation (N-glycosylations at Asn272, 319, and 353 are particularly important for proper maturation and stability), disulphide bond formation, and chaperone binding for proper folding. Tyrosinase first acquires enzymatic activity in the TGN, likely because this is where it first encounters its critical cofactor, the copper cation [96,183,184].

Tyrosinase catalyses the hydroxylation of L-Tyr to L-DOPA (tyrosine hydroxylase activity), the oxidation of DOPA to DQ (DOPA oxidase activity), and the oxidation DHI and DHICA into the eumelanin precursors indole-5,6-quinone and indole-5,6-quinone carboxylic acid, respectively. Tyrosinase is activated by its own substrate L-Tyr and co-factor L-DOPA; the latter accelerates the conversion of tyrosine into DQ by removing an initial lag period observed for tyrosine hydroxylation. On the contrary, DHI can inhibit its tyrosine hydroxylase activity and tyrosine can inhibit the conversion of DHI to indole-5,6-quinone. Tyrosinase exhibits optimal enzymatic activity at neutral pH and it is suppressed at pH below 5.8 [183,184,185,186].

##### Tyrosinase-Related Protein 1 (TYRP1)

Although TYRP1 has an undisputed role in melanin synthesis (mutations in the encoding gene cause a form of oculocutaneous albinism, OCA3), its catalytic activity remains unclear. From the several enzymatic activities attributed to the mouse homologue (tyrosine hydroxylase, L-DOPA oxidase, dopachrome tautomerase, DHICA oxidase), the human TYRP1 only appears to display tyrosine hydroxylase activity. However, it has been implicated in the maturation and stabilization of tyrosinase [184,187,188].

##### Cystine Transporters

The synthesis of pheomelanin is dependent on the availability of cysteine, obtained through redox exchange of cystine with GSH. SLC7A11 is a cystine/glutamate exchanger, responsible for cystine transport into melanocytes and possibly into melanosomes; additionally, it also seems to regulate the transcription of several melanogenesis-related genes [189,190]. Cystinosin (CTNS) is a cystine/H^+^ symporter located in melanosomes that controls the efflux of cystine; its ability to efflux protons along with cystine also renders it as having a role in the control of melanosomal pH [191].

#### 3.2.2. Control of Melanosomal pH

During the early stages of maturation (I and II), the melanosomal pH is low (around pH 4), which seems to favour the formation of PMEL fibrils [192]. However, in later stages, the activity of tyrosinase requires an increase in the pH. At a weakly acidic pH (5.8–6.3), the production of pheomelanin is preferred because CD-quinone cyclization (bicyclic intermediate of pheomelanin) proceeds much faster than the cyclization of DQ (first step of eumelanogenesis). At near neutral pH (6.8–7.4), eumelanin production is favoured [165,193,194,195].

Since the pH of melanosomes is greatly influenced by their internal ionic equilibrium, it has been suggested that pH control can be achieved through the concerted action of a cascade of ion channels (that allow ions to diffuse down their electrochemical gradient) and transporters (that actively transport ions against electrochemical gradients)—Figure 6 [15]. In turn, recently, it was proposed that soluble adenylyl cyclases (sACs) can modulate the melanosomal pH and pigmentation, likely via the regulation of such ion transporters through the stimulation of EPAC (exchange of proteins activated by cAMP) [196].

Vacuolar ATPases (V-ATPases) are known to be located in melanosomes, being a key component in melanosome acidification [121,194,197,198]. To increase the pH, the protons taken up by V-ATPases are returned to the cytoplasm through sodium–proton exchangers. Melanocytes express cytoplasmic membrane sodium hydrogen exchangers (NHEs, namely NHE-3 and NHE-7) that appear to be involved in this process [199]. More important, OCA2 and SLC45A2 have been recognized as major players in regulating melanosomal pH, and as NHEs, they also have been proposed to act as sodium–proton exchangers and proton/glucose exporters, respectively [15].

OCA2 (also known as P protein) is a melanosomal membrane protein with a predicted structure consisting of 12 transmembrane domains [121,200,201]. Besides acting as a sodium–proton exchanger, OCA2 has been suggested to be an anion channel, making melanosomes less acidic by allowing chloride ions to flow out of the melanosome and thus modulating the membrane potential with consequent reduced vacuolar H^+^-ATPase activity [202]. Interestingly, another study found that the mere expression of chloride voltage-gated channel 7 (CLC7) is implicated in the pigmentation and more recently, in a non-peer reviewed report, with counteracting effects to OCA2 [203,204]. The efflux of Cl^-^ ions by itself is also important in the context of melanogenesis because they were found to inhibit human tyrosinase activity; in the same work, the Cl^-^ transporters SLC12A4 and SLC12A7 were found to be downregulated during TYR-mediated melanogenesis in HEK293 cells, providing further evidence of the Cl^-^-mediated negative regulation of melanogenesis [205]. SCL45A2 (also known as membrane-associated transporter protein or MATP) is a melanosomal transporter, with a predicted structure consisting of 12 transmembrane domains, sharing homology with sucrose/proton symporters. As OCA2, besides being involved in the regulation of pH and consequent tyrosinase activity, SCL45A2 has been implicated in tyrosinase trafficking and also in the melanosome structure [121,197,206,207].

As the removal of H^+^ from melanosomes implies the exchange with Na^+^, an excess of these ions needs to be cycled out of the melanosomes. SLC24A5 (also known as sodium/potassium/calcium exchanger 5 or NCKX5) has been detected in melanosome membranes, and its K^+^-dependent Na^+^/Ca^2+^ exchanger activity was already demonstrated to control melanogenesis [121,208,209]. The exchanger activity of SLC24A5 also provides a link between cytosolic and melanosomal Ca^2+^; melanosomes are enriched in Ca^2+^ because melanin has the ability to bind and chelate it, which has been implicated in the regulation of calcium homeostasis in melanocytes [210,211]. Additionally, the transport of Ca^2+^ (and other ions) across the melanosomal membrane likely regulates its voltage, facilitating the fusion process between melanosomes and vesicles of protein transport or other organelles; in this context, melanosomes were found to form mitofusin 2-dependent contacts with mitochondria, which have implications in melanosome maturation and melanin synthesis [121,212]. In fact, mitofusin 2 was established as a negative regulator of melanogenesis [213]. It is important to mention that the melanogenic effect of Ca^2+^ is not restricted to the actions upon the melanosomal membrane. Cytosolic Ca^2+^ is known to activate protein kinase C (PKC), which modulates the pigment levels in melanocytes [214,215,216]. Ca^2+^ is also required for the transport of L-phenylalanine into melanocytes (calmodulin-dependent Ca^2+^-ATPase) and for its turnover into L-Tyr [82,217]. Recently, the small conductance Ca^2+^-activated K^+^ channels were also suggested to be functionally important in human pigmentation, although the mechanism of action needs further elucidation [218]. Finally, Ca^2+^ contributes to the process of melanosome transfer between melanocytes and keratinocytes [219]. In melanocytes, some ion channels at the plasma membrane have already been implicated in the Ca^2+^ influx and modulation of pigmentation: melanocyte-specific transient receptor potential cation channel, subfamily M, member 1 (TRPM1) [220,221,222], transient receptor potential cation channel, mucolipin subfamily, member 3 (TRPML3) [223], and calcium release activated channels (CRAC) [224], among others [121].

The concerted neutralization of melanosomal pH downstream of the activity of SLC24A5 was thought to be regulated by two pore segment channel 2 (TPC2) in virtue of the suspected activity in mediating the nicotinic acid adenine dinucleotide phosphate (NAADP)-dependent efflux of Ca^2+^ [15,225]. However, TPC2 was later found to function as an Na^+^-selective channel [226]. Moreover, it was uncovered that TPC2 provides the negative regulation of melanogenesis by increasing melanosomal membrane potential and acidity, possibly by providing a cation counterflow to enhance the H^+^ transport by V-ATPases [227]. Thus, although TPC2 does not contribute to the efflux of Ca^2+^, its activity seems to be part of the normal regulation of proton flux that maintains ion homeostasis, and it may be important in melanosome acidification needed in early stages of melanosome maturation. In any case, other ion channels or transporters must exist in the melanosome membrane that counterbalance the influx of Ca^2+^. Besides its location in the plasma membrane, TRPML3 is also found in endosomal compartments, and if present in melanosomes, it may mediate this transport [121]. Recently, the knockdown of *Mgrn1* in a mouse melanoma cell line, which encodes mahogunin ring finger 1 (MGRN1), a RING-finger nuclear-cytoplasmic E3-ubiquitin ligase ubiquitously expressed in mammals, was associated with increased TYR specific activity and pigment production due to the neutralization of melanosomal pH, through the induction of the Ca^2+^ transporter mucolipin transient receptor potential cation channel 3 (MCOLN3) expression, also termed TRPML3 [228].

### 3.3. Transport and Transfer of Melanosomes

To ensure pigmentation of the hair shafts, melanocytes transfer mature melanosomes into neighbouring keratinocytes via dendritic projections. However, before being transferred to the keratinocytes, melanosomes need to be transported from their site of origin at the perinuclear area to the melanocyte periphery. The motility of melanosomes in melanocytes occurs along microtubules (composed of α:β-tubulin dimers) and actin filaments (composed of actin monomers)—Figure 7 [229].

Microtubules act as tracks for the transport of melanosomes to the peripheral region. The movement along microtubules is influenced by two classes of microtubule-associated motor proteins, the kinesins and the dyneins [85,230,231]. A complex comprising RAB1A, skeletal muscle and kidney enriched inositol phosphatase (SKIP)/pleckstrin homology domain-containing, family M, member 2 (PLEKHM2) and kinesin-1 motor [composed of two KIF5B heavy chains and two kinesin light chain 2 (KLC2)] has been involved in the anterograde transport of melanosomes [232,233]. The Rab-interacting lysosomal protein (RILP), the p150Glued subunit of the dynein–dynactin motor complex (also known as dynactin subunit 1 or DCTN1), melanoregulin, and RAB36 are reported to be involved in the retrograde movement of melanosomes (Figure 7) [234,235].

At the periphery of melanocytes, the capture and movement of melanosomes in the rich network of actin filaments of the dendrites is regulated by the RAB27A–melanophilin(MLPH)–MYO5A complex [236,237,238,239,240,241,242]. RAB27A is responsible for linking synaptotagmin-like 2 (SYTL2) with phosphatidylserine, docking melanosomes at the plasma membrane [78,85,240,243,244]. MLPH, previously known as SLAC2-A, is an effector of RAB27A, and it functions as a linker between RAB27A and MYO5A; prohibitin (PHB) is necessary for the interaction between MLPH and RAB27A [239,245,246,247]. MYO5A is an ATP-dependent motor protein that captures melanosomes in subcortical bundles at the periphery of dendrites and moves the melanosomes towards the plus end of actin filaments [238,248].

#### Models of Melanin Transfer

Once melanosomes are docked to the inner face of the melanocyte membrane, melanin is transferred to keratinocytes. The exact mechanism is still unclear; several models are still currently proposed for the transfer of melanin from melanocytes to keratinocytes, based on different cellular models and methodologies used by different research groups (Figure 7) [249,250]. Recent in vitro studies have provided strong evidence for the coupled mechanism exocytosis/endocytosis underlying epidermal melanin transfer between melanocytes and keratinocytes. The authors proposed that basal melanin transfer occurs through phagocytosis of melanocores (melanin core of melanosomes without the delimiting membranes), while under stress conditions, such during tanning, macropinocytosis of melanosome-laden globules seems to be the preferred mechanism [251,252].

The exocytic-mediated transfer model states that melanosomal membrane fuses with the melanocyte membrane, resulting in the extracellular release of the melanin core, known as the melanocore. The pigment is posteriorly engulfed by keratinocytes through an endocytic process, probably phagocytosis, acquiring a new delimiting membrane, originated from the plasma membrane of keratinocytes; these organelles were named melanokerosomes [253]. This theory is supported by electron microscopic observations of human skin and hair follicles depicting uncoated melanin in the intercellular space [254,255,256]. Further support is provided by the fact melanocytes express several SNAREs and Rab GTPases with well-known roles in the regulation of exocytosis, and some of them have even been implicated in the melanin transfer process [249,255]. In addition, by sharing an embryonic origin, melanocytes are closely related to neural cells, of which synaptic vesicles are known to undergo exocytosis [249]. Rachinger et al. [257] demonstrated that alpha-synuclein, a presynaptic protein, is expressed in melanocytes and that it is involved in melanosome trafficking and release processes, in vitro. Recently, the requirement of the exocyst tethering complex in melanin transfer to keratinocytes, involving RAB11B, exocyst complex component 4 (EXO4, previously SEC8), and EXO70, was reported [258].

The cytophagocytic-mediated transfer model proposes that melanin is transferred to keratinocytes by phagocytosis of the tip of a melanocyte dendrite or by the phagocytosis of filopodia that protrude from the dendrites (Figure 7). This would imply that internalized pigments within keratinocytes would be surrounded by three membranes from different origins [259]. This was supported mainly by electron and time-lapse video microscopy studies from a time when less advanced techniques were used, with later optimized acquisition techniques failing to corroborate it [249]. The enclosure of filopodia by keratinocytes was suggested more recently as a filopodia–phagocytosis model for melanin transfer. This model proposes that MYO10 drives the formation and elongation of filopodia in melanocytes. Then, via integrins, melanocytes adhere to the membrane of keratinocytes and the MYO10-associcated motor force at the filopodia tips helps filopodia insertion into the plasma membrane of keratinocytes, causing their uptake [260]. Interestingly, in this context, filopodia could also fuse to the keratinocyte membrane and be a variation of the fusion model for melanin transfer to keratinocytes [253].

The fusion model assumes that melanocytes and keratinocytes fuse their membranes to connect their cytoplasms (Figure 7). This fusion appears to involve filopodia that extend from the dendrite of melanocytes and adhere to keratinocytes, resulting in the formation of conduits that permit unidirectional transport of melanosomes from melanocytes to keratinocytes [261]. Although this kind of organelle transport exists in other cells and the transfer of melanosomes at filopodia locations has been observed, definitive proof of membrane fusion and melanosome transport across the putative formed channels has yet to be given [249].

The membrane vesicle-mediated transfer model reports that melanosome-containing vesicles are released into the extracellular space and adhere to keratinocytes, and then, they are internalized via phagocytosis [249,250]. It has also been hinted that multiple melanosomes can be concentrated at the filopodia to be released inside globules from various areas of the dendrites [262,263].

Independently of the particular transfer mechanism, the filipodia structures on melanocyte dendrites seem to be important for pigment transfer [264,265]. Rab17 siRNA knockdown in melanoma cells inhibited filopodia formation and led to a quantitative increase in the melanosome concentration at the cell periphery. Rab17 knockdown did not inhibit melanosome maturation or movement, but it caused the accumulation of melanin inside melanoma cells [265].

Although the exact mechanism or a combination of mechanisms of melanin transfer remain unclear, phagocytosis (a process of cellular engulfment of particles with a diameter of more than 0.5 μM) is a mostly necessary step, and it has been reported to be modulated mainly by the protease-activated receptor 2 (PAR2, also known as coagulation factor II receptor-like 1 or F2RL1) [266,267,268,269,270,271]. PAR2 is a seven-transmembrane G protein-coupled receptor, related to the thrombin receptors. The proteolytic cleavage at the extracellular N-terminal moiety exposes a new N-terminus that acts as a tethered ligand and binds the receptor leading to its activation. This activation induces the secretion of serine proteases by keratinocytes that are responsible in the first place for the cleavage of PAR2, thus creating a positive feedback loop [266,267]. Even though PAR2 performs the lead role in phagocytosis associated with melanin transfer, it is not expected to be the only receptor that modulates the process as suggested by the incomplete inhibition of melanin transfer upon treatment with serine protease inhibitors. It is important to mention is that the modulation of pigmentation by PAR2 goes beyond the direct activation of the phagocytic pathway in keratinocytes: PAR2 activation also increases melanocyte dendricity (upon stimulation of prostaglandins PGE2 and PGF2α released by keratinocytes) and the expression of SCF, a paracrine regulator of melanogenesis [272,273]. Transient receptor potential cation channel subfamily A member 1 (TRPA1) is a calcium-permeant, non-selective, cation channel that has been implicated in the cellular sensing of physical and chemical stimuli, such as UV radiation, heat shock, and oxidative stress. Recently, studies have identified TRPA1 as an inducer of PAR2 expression and the activation in keratinocytes [274]. It would be important to confirm that what we know about melanin transfer from research on the epidermal pigmentary unit also applies to melanin transfer in the follicular pigmentary unit.

## 4. Regulation of Follicular Melanogenesis

Melanogenesis is under complex control involving several positive and negative regulators/factors that act via autocrine, paracrine, and endocrine mechanisms (Figure 8). Although much of the available data regarding the control of melanogenesis pertain to human epidermal or murine follicular melanocytes, it is thought that such regulators of epidermal melanogenesis contribute in a similar way to human hair pigmentation. One possible exception is UV radiation, the principal regulator of melanin synthesis in the epidermis [50,85]. Ignoring the possible shadow effect provided by hair on skin zones where hair density is high, like the scalp, in general and in “naked skin”, radiation in the UV range is not expected to reach more than 160 μm below the skin surface (into the dermis) [275,276]. Since hair bulbs from terminal hairs are located between 1 and 5 mm below the surface, in the subcutaneous fat, active follicular melanocytes are more protected from UV direct stimulation than epidermal melanocytes [277].

### 4.1. Pro-Opiomelanocortin (POMC)-Derived Peptides

The hair follicles are local sources and targets for pro-opiomelanocortin (POMC)-derived peptides; although originally discovered in the anterior pituitary, these peptides are also expressed and secreted by melanocytes and keratinocytes [50,85,278,279]. POMC transcription and translation is hair cycle-dependent and increases significantly in the anagen phase. The proteolytic processing of POMC by PC1 produces the peptides adrenocorticotropin (ACTH) and β-Lipotropin (β-LPH), with the latter being further processed to β-endorphin (β-END) and β-melanocyte-stimulating hormone (β-MSH). PC2 cleaves the first 14 amino acids of the ACTH sequence to generate ACTH(1–14)OH, the precursor of α-MSH [280,281,282]. POMC-derived products are the main regulators of follicular melanogenesis, and even unprocessed POMC was reported to stimulate melanin production (Figure 8). Additionally, POMC-derived peptides also influence hair growth [279,283,284,285,286,287].

ACTH, α-MSH, and β-MSH share an essential core primary sequence of four amino acids (His-Phe-Arg-Trp) that allows the binding to melanocortin receptors. While the latter acts on melanocortin 4 receptor (MC4R), ACTH and α-MSH bind to MC1R, a G protein-coupled membrane receptor abundantly expressed in melanocytes of hair follicles. Nonetheless, the activation of both receptors leads to the stimulation of adenylyl cyclase (AC), an enzyme that catalyses the conversion of ATP to cAMP (cyclic adenosine monophosphate) [50,85,288,289,290,291]. Then, cAMP binds to the regulatory subunit of protein kinase A (PKA), allowing the catalytic subunit to be liberated and activated; the termination of this cellular signalling is provided by phosphodiesterases (PDEs), which hydrolyse free and PKA-bounded cAMP, leading to reassociation of the two subunits [292]. Activated PKA is translocated to the nucleus where it induces the phosphorylation of CREB at Ser133. Phosphorylated CREB activates the expression of specific genes containing the consensus cAMP responsive element (CRE) sequences in their promoters, such as *MITF*; PKA also phosphorylates the nuclear transcriptional coactivator CBP, which interacts with CREB-family proteins for PKA-dependent gene expression. Consequently, the increased transcription of *MITF* upregulates the expression of several genes involved in synthesis of melanin, but also in the regulation of melanogenesis (as *MC1R*), melanocyte proliferation, and dendrite formation [50,85,289,293,294]. Besides the direct phosphorylation of CREB, PKA also increases the transcription of *MITF* by inhibiting (via phosphorylation at Ser587) salt-inducible kinase 2 (SIK2) activity. SIK2 promotes the phosphorylation of CREB-regulated transcription coactivator 1 (CRTC1), preventing its translocation from the cytoplasm to the nucleus, an essential step for CREB-mediated gene expression. CRTC1 and SIK-2 have been shown to be fundamental determinants of the melanogenic program in mice [295,296].

The peroxisome proliferator-activated receptor γ coactivator-1α (PGC-1α) and PGC-1β are other transcriptional regulators that are modulated by the cAMP/PKA pathway. α-MSH signalling increases the expression of PGC-1α and stabilizes both PGC-1α and PGC-1β proteins, probably through phosphorylation mediated by PKA, with the consequent stimulation of MITF, tyrosinase expression, and melanin synthesis [297]. PGC-1α and PGC-1β coactivate many transcription factors, including the nuclear hormone peroxisome proliferator-activated receptor γ (PPARγ), which has also been shown to be involved in Ꮁ-MSH-induced melanogenesis; other PPAR subtypes were reported to be expressed in melanocytes, and PPARα, but not PPARβ/δ, is known to contribute to the control of melanogenesis [298,299,300,301].

PKA also inhibits the activity of phosphatidylinositol 3-kinase (PI3K) and one of its key effectors, the serine/threonine kinase Akt; the activation of Akt depends on the phosphorylation of its Thr308 and Ser473 residues upon binding to the PI3K phospholipid products. Inactive Akt is incapable of phosphorylating glycogen synthase kinase 3β (GSK3β) at Ser9, which promotes its inactivation. Thus, GSK3β can phosphorylate MITF at Ser298, which facilitates its binding to the *TYR* promotor and stimulates melanogenesis. Curiously, PKA can promote the degradation of GSK3β (phosphorylation at Ser9) via cross-talk with the Wnt/β-catenin signalling pathway, addressed below [302,303,304,305]. The inhibition of the PI3K/Akt pathway by PKA not only avoids the degradation of GSK3β, but also prevents activation of the serine/threonine kinase p70S6K1 and mammalian target of rapamycin (mTOR) along with its complex 1 (mTORC1). Both factors have been implicated in the negative regulation of melanogenesis, and cAMP-induced melanogenesis has shown to be, at least in part, mediated by their inhibition [306,307,308,309,310,311].

The remaining POMC-derived peptide, β-END, has been shown to positively regulate melanogenesis, proliferation, and dendricity in melanocytes of the epidermis and hair follicles by binding to its μ-opiate receptor. Contrarily to other POMC-derived peptides, β-END is not expected to exert its melanogenic modulatory effect through the cAMP/PKA signalling pathway; in fact, signalling through μ-opiate downregulates the level of cAMP. A PKC-dependent pathway (addressed bellow) has been suggested as a downstream mechanism for the stimulation of melanogenesis by β-END [50,85,283,312].

#### 4.1.1. Corticotropin-Releasing Factor (CRF)

Corticotropin-releasing factor (CRF, also known as corticotropin-releasing hormone or CRH) is the most proximal element of the hypothalamic–pituitary–adrenal axis (HPA). The binding of CRF to its receptors in the skin mediates POMC expression in melanocytes, keratinocytes, and Langerhans cells and subsequent production of the derived peptides [85]. Both skin and hair follicles contain an equivalent to the HPA axis, with keratinocytes and melanocytes expressing CRF and the corresponding G protein-coupled corticotropin-releasing factor receptors (CRFR); isoforms of the CRF1R subtype are expressed in the epidermis, while isoforms of the CRF2R subtype are expressed in the hair follicle. CRF can induce the synthesis of POMC through signalling involving PKA and PKC (Figure 8) [50,85,313,314].

#### 4.1.2. Thyrotropin-Releasing Hormone (TRH)

Thyrotropin-releasing hormone (TRH) is the most proximal regulatory element of the hypothalamic–pituitary–thyroid axis (HPT), controlling the production of thyroid hormone. TRH is recognized as a potent stimulator of POMC expression, with human skin and hair follicles being reported to transcribe it along with its receptor (TRHR). The expression of TRH stimulates growth, melanin synthesis, and melanocyte dendricity in hair follicles; in the epidermis, TRH does not appear to affect melanogenesis. In hair follicles, the expression of TRHR is confined to the inner root sheath, being absent in the follicular pigmentary unit. This suggests that THR stimulates the production of POMC in keratinocytes, with the derived peptides posteriorly acting on melanocortin receptors of melanocytes (Figure 8). Alternatively, TRH have also been proposed to bind directly to MC1R, which could explain the reported cases of normal hair pigmentation in the absence of melanocortin synthesis [315,316]. Besides TRH, the related thyroid hormones have also been shown to directly affect human hair pigmentation [317].

### 4.2. WNT Proteins

The Wnt family consist of cysteine-rich lipoglycoprotein members that act on G protein-coupled frizzled (FZD) receptors. The binding to FDZ receptors activates an intracellular cascade of events involving (canonical form) or not involving (noncanonical form) the transcription regulator β-catenin [85].

The non-canonical pathways are diverse but poorly characterized; melanin suppression by WNT5 is known to be dependent on the noncanonical Wnt/Ror2 pathway [318]. In the canonical pathway, the binding of WNTs to FZD receptors and to the low-density lipoprotein-related protein (LRP) 5/6 coreceptor inhibits the constitutive degradation of β-catenin by GSK3β, promoting its accumulation; further stability of β-catenin is provided by PKA through the phosphorylation of Ser675 (Figure 8) [305,319]. β-Catenin is then translocated into the nucleus, where it interacts with T-cell factor (TCF)/LEF transcription factors to upregulate the transcription of *MITF*. β-Catenin also interacts directly with MITF (phosphorylating the Ser298 residue) and increases the binding of MITF to the M box of the *TYR* promotor [85,320,321].

In the absence of WNTs, β-catenin is phosphorylated (Ser33, Ser37, Ser45, and Thr41) by a multiprotein complex containing axin, adenomatous polyposis coli (APC), casein kinase Iα (CKIα), and GSK3β; these phosphorylations target β-catenin for ubiquitination and proteasomal degradation [305,322,323]. The activation of the Wnt/β-catenin signalling pathway by WNT1, WNT3a, WNT7A/B, or WNT10B has been implicated in the upregulation of melanogenesis, melanocyte differentiation and proliferation, and hair follicle regeneration [324,325,326,327,328,329]. Many other ligands have also been shown to modulate the Wnt signalling pathway and the consequent production of melanin by interacting with WNT proteins: Dickkopf 1 (DDK1) [330,331], WNT inhibitory factor 1 (WIF1) [332], and secreted frizzled-related protein 2 (SFRP2) [333].

### 4.3. Stem Cell Factor (SCF)

SCF is a growth factor secreted by keratinocytes and fibroblasts. SCF is the specific ligand of KIT, which is expressed in several types of cells, including melanocytes. The binding of SCF to the extracellular cellular domain of KIT prompts its dimerization, activation of intrinsic tyrosine kinase activity, and autophosphorylation. Then, KIT can phosphorylate various substrates that lead to the activation of the mitogen-activated protein kinase (MAPK) signalling pathways and to the modulation of melanocyte proliferation, differentiation, and melanogenesis. Of note, SCF/KIT signalling can also activate PI3K, for which, effects on melanogenesis were discussed before, in Section 4.1 [50,85,334,335,336,337].

There are three well-characterized subfamilies of the MAPK superfamily known to have crucial roles in melanin synthesis: the extracellular signal-regulated kinases (ERKs, mainly 1/2), the c-Jun N-terminal kinases (JNKs, mainly 1/2, also known as stress-activated protein kinases or SAPKs), and p38-MAPK (Figure 8) [85,338,339]. The activation of ERK, which can additionally be achieved through cAMP/PKA or PKC-mediated signalling, is known to cause the phosphorylation of MITF at Ser73, allowing the recruitment of the transcriptional coactivator CBP/P300 of CREB and consequent transcription of melanogenic enzymes. However, such phosphorylation, along with another one at Ser409, induced by p90 ribosomal S6 kinase (RSK, ERK downstream activation), also leads to MITF destabilization via increased ubiquitination, followed by proteasome-dependent degradation; in due course, the production of melanin is decreased [85,87,340,341,342,343,344,345,346]. Several lipidic second messengers have been shown to modulate melanogenesis, mainly through the ERK signalling pathway: sphingosylphosphorylcholine [309,347,348,349,350], ceramide [351,352,353], sphingosine-1-phosphate [354,355], and lysophosphatidic acid [356]. The activation of JNK has also been reported to inhibit melanogenesis via the degradation of MITF. In addition, the antimelanogenic effect of JNK was linked to the blockage of the nuclear translocation of CRTC3 (CREB-regulated transcription coactivator 3) and further impairment of the transcriptional activity of CREB [357]. The activation of p38-MAPK induces the phosphorylation of CREB, via downstream MSK1 (mitogen- and stress-activated protein kinase 1), activating MITF, which promotes the transcription of melanogenesis-related genes and the increase in melanin synthesis. However, there is also some evidence demonstrating the involvement of p38-MAPK in the inhibition of melanin production via the proteasome-dependent degradation of tyrosinase and other related proteins [337,358,359,360].

### 4.4. Endothelin 1 (ET-1)

Endothelins (ETs) are peptides present in a broad range of tissues, including the skin. The three known ET isopeptides (ET-1, 2, and 3) are obtained through the cleavage of preproendothelin by prohormone convertases. ET-1 is synthesized and secreted by keratinocytes, acting on the G protein-coupled receptor EDNRB of melanocytes with the ensuing stimulation of proliferation, melanogenesis, and dendricity (Figure 8) [50,85,361,362,363,364,365].

Upon the binding of ET-1 to EDNRB, phospholipase Cγ (PLCγ) is activated, increasing the hydrolysis of phosphatidylinositol 4,5-bisphosphate (PIP_2_), with the further generation of inositol–triphosphate (IP_3_) and diacylglycerol (DAG). IP_3_ raises the intracellular concentration of Ca^2+^, and DAG induces the activation of Ca^2+^/phospholipid-dependent PKC. Subsequently, PKC activates the MAPK signalling pathway, which culminates in the phosphorylation of CREB and induction of *MITF* transcription [50,85,224,361,366]. PKC is also known to be translocated from the cytosol to the melanosomal membrane (upon interacting with receptor for activated C-kinase 1, RACK1), promoting the phosphorylation of Ser505 and Ser509 on the cytoplasmic domain of tyrosinase; such phosphorylation increases the activity of the enzyme and induces the formation of a complex with TYRP1, which further enhances melanin synthesis [214,215,216,367].

### 4.5. Neurotransmitters

Acetylcholine (ACh) is synthesized and degraded by keratinocytes and melanocytes; the degradation process is controlled through the action of acetylcholinesterases (AChEs). ACh interacts with G protein-coupled muscarinic receptors, with all five subtypes (M1R-M5R) being detected in human melanocytes. The activation of M1R, M3R, and M5R stimulates phosphoinositide metabolism, whereas M2R and M4R hinder the activity of AC (Figure 8). Both events are known to inhibit the synthesis of melanogenesis, and the control of melanogenesis by ACh has been proposed by several researchers [50,368,369,370,371].

Glutamate is supplied to melanocytes by adjacent keratinocytes. The activation of G protein-coupled metabotropic glutamate receptor 6 (mGluR6) increases melanin production via the activation of the TRPM1 calcium channel (Figure 8). Glutamatergic signalling through ionotropic glutamate receptors [glutamate receptor 2 (GluR2), glutamate receptor 4 (GluR4), α-amino-3-hydroxy-5-methyl-4-isoxazolepropionic acid (AMPA) receptor, and N-methyl-D-aspartate receptors 2A and 2C (NMDAR2A, NMDAR2C)] is implicated in the transcriptional regulation of *MITF* (Figure 8) [85,222,372].

Histamine generation has been demonstrated in keratinocytes, being present in several skin compartments, including hair follicles. Although melanocytes express several histamine receptors, the increased production of melanin induced by histamine, appears to be related to the stimulation of the G protein-coupled H_2_ receptor, which causes cAMP accumulation and subsequent PKA activation (Figure 8) [50,373,374,375]. Despite that, a depigmenting effect of histamine has also been demonstrated [376,377].

Norepinephrine and epinephrine act on G protein-coupled α_1_ and β_2_ adrenergic receptors of melanocytes. Autocrine and paracrine (keratinocytes) norepinephrine are the preferred ligand for the α_1_ adrenoreceptor, stimulating melanogenesis via a PKC-dependent pathway (Figure 8). Epinephrine (produced by keratinocytes) preferentially targets the β_2_ adrenergic receptors, stimulating melanin through AC and the downstream cAMP/PKA signalling pathway (Figure 8) [50,82,85,93,378].

Serotonin (5-hydroxytryptamine, 5-HT) is produced and metabolized by melanocytes and keratinocytes. The actions of 5-HT are mediated by its interaction with G protein-coupled receptors (5-HTR), which are widely detected in melanocytes. 5-HT was shown to positively modulate, in vitro (cell culture and hair follicles) and in vivo (mice and zebrafish), the production of melanin through interactions with 5-HT_1A_R, 5-HT_1B_R, and 5-HT_2A_R (Figure 8) [50,379,380,381]. However, recently, it was found that the interaction between neurokinin 1 receptor (NK1R) and 5-HT_1A_R prevents the synthesis of melanin induced by the activity of the latter [382]. NK1R is activated by Substance P (SP), an undecapeptide belonging to the tachykinin family, causing a decrease in melanin production by inhibiting the expression of MITF and p-p38-MAPK and increasing p-p70S6K1 [382,383,384]. Moreover, SP can also inhibit the expression of 5-HT_2A_R and neutralize the pro-melanogenic effect of 5-HT [385]. Paradoxically, NK1R and SP have been reported to positively regulate melanogenesis through the Wnt/β-catenin signalling pathway and increased secretion of ET-1 [330,386]. Melatonin (MTN), obtained from 5-HT, was also reported to decrease the production of melanin. The effects of MTN are mediated by the binding to nuclear or membrane-bound G protein-coupled MTN receptors (MTNR1A and MTNR1B), both found on melanocytes. The activation of membrane-bound melatonin receptors is known to inhibit AC (Figure 8) [50,387,388].

### 4.6. Bone Morphogenetic Proteins

BMPs are the largest family of secreted signalling molecules of the transforming growth factor (TGF)-β superfamily. They are powerful regulators of skin development, controlling epidermal homeostasis, hair follicle growth, and skin/hair pigmentation. BMP signalling is influenced by the BMP type, presence of antagonists, development stage of the target tissue, and target cell receptors. Keratinocytes and melanocytes express BMP4 and BMP6, their receptors (serine/threonine kinase receptors type I and II, BMPR1 and BMPR2), and the respective antagonists, sclerostin and noggin, altogether affecting the production of melanin (Figure 8) [50,389,390,391,392]. BMP6 has been shown to stimulate melanogenesis by upregulating tyrosinase expression and activity; additionally, it also stimulates the formation of filopodia and MYO10 expression, associated with increased melanosome transfer from melanocytes to keratinocytes. The effects of BMP6 are achieved through activation of the canonical BMP/Smad pathway, and non-canonical p38-MAPK and PI3K pathways; the canonical BMP/Smad pathway involves the phosphorylation of cytosolic transcription regulators (Smad proteins), which are translocated to the nucleus where they can alter gene expression [50,389]. On the contrary, BMP4 downregulates melanogenesis by targeting MITF for proteasome-mediated degradation, via the MAPK/ERK pathway [85,389,391,392,393]. TGF-β1, another member of the TGF-β superfamily, also provides negative regulation of melanogenesis via the MAPK/ERK pathway [394,395,396,397,398].

### 4.7. Oestrogens

Oestrogens have significant effects on many aspects of skin physiology, including hair growth and pigmentation. The binding of oestrogens to their receptors upregulates MITF and TYR and consequently melanin contents, with the involvement of the PKA pathway (Figure 8) [50,366,399,400].

Classically, the oestrogen action is viewed as mediated by two nuclear oestrogen receptors: ERα and ERβ; keratinocytes and melanocytes express both receptors, with the latter being preferentially expressed in the hair follicles [50,366]. More recently, a G protein-coupled oestrogen receptor (GPER) has also been reported. The activation of this receptor also enhances melanogenesis via the cAMP/PKA signalling pathway (Figure 8) [401,402].

### 4.8. Cell Adhesion Molecules

P-cadherin, which regulates cell–cell recognition and signalling, is involved in hair follicle morphogenesis, hair fibre production, and follicular regression; additionally, it was found to be a regulator of pigmentation. The silencing of P-cadherin in organ cultures of melanogenically active human scalp hair follicles significantly reduced melanogenesis, possibly via GSK3β-mediated Wnt signalling. Interestingly, the pigmentary role of P-cadherin seems to be exclusive to hair follicles, as epidermal pigmentation was unaffected by its knockdown in organ-cultured skin [403]. The cadherin desmoglein 1 (DSG1), which regulates keratinocyte-specific cell–cell adhesion, was also found to regulate keratinocyte–melanocyte paracrine crosstalk and inhibit the production of melanin [404]. Neuregulins are a group of peptide growth factors that mediate cell–cell interactions in several organ systems through tyrosine kinase receptors of the ErbB family. Neuregulin 1 (NRG1), secreted by fibroblasts, was described as increasing the pigmentation of melanocytes in tissue culture and in an artificial skin model. [405]. Laminin-332, derived from keratinocytes, not only plays a critical role in adhesion-related cell functions in melanocytes but it also regulates melanogenesis by controlling the uptake of tyrosine [406].

### 4.9. Non-Coding RNAs

Non-coding RNAs (ncRNAs), particularly microRNAs (miRNAs) and the long non-coding RNAs (lncRNAs), are emerging as new regulators of the pigmentation process. miRNAs are about 20–25 nucleotides in length that, by targeting specific mRNAs, cause their degradation or inhibit their translation into proteins [85]. Several miRNAs have been shown to interfere with pigmentation by targeting *MITF* mRNA: miR-25, miR-137, miR-141-3p, miR-197, miR-200a-3p miR-218, miR-328, miR-574-5p, miR-634, miR-675, miR-720, miR-766, miR-1225-3p, miR-1308, and miR-3196 [407,408,409,410,411,412,413,414]. Many miRNAs target mRNAs other than the *MITF* transcript: miR-21a-5p (*SOX5*), miR-27a-3p (*WNT3A*), miR-125b (SRC homology 3 domain-binding protein 4, *SH3BP4*), miR-137 (*KIT*), miRNA-143-5p (*MYO5A*), miR-145 (*SOX9*, *TYR*, *TYRP1*, *MYO5A*, *RAB27A*, *FSCN1*, but also *MITF*), miR203 (*KIF5B*, and possibly *CREB1*), miR-211 (TGFβ receptor 2, *TGFBR2*), and miR-434 (*TYR*) [411,415,416,417,418,419,420,421,422,423,424,425]. Many regulators of pigmentation, with unknown targets, have also been disclosed: miR-9, miR-21, miR-28, miR-130b, miR-139-5p, miR-155, miR-182, miR-193, miR-206, miR-221, miR-222, miR330-5p, miR-335, miR-365, and miR-455 [422,426,427,428]. Of note, miR-200c, miR-230, miR330-5p, miR-675, and miR-3196 are not produced by melanocytes, being delivered from keratinocytes via exosomes [411,412,427,429]. LncRNAs are greater than 200 bp in length and regulate diverse biological functions and also some microRNAs [85]. Some lncRNAs (CD1C-2:1, H19, SRA, TCONS_00049140, UCA1) have already been shown to have an important role in melanogenesis, although their role in not yet fully described [430,431,432,433].

### 4.10. Other Regulators

Zinc α-2-glycoprotein is produced by keratinocytes, and it was proposed to play a part in the negative regulation of melanin production [434]. An undisclosed member of the heat shock protein (HSP) 70 family has been shown to suppress melanin production in vitro and in vivo through direct interactions with MITF; HSPs are constitutively expressed in the skin and confer protection against stressors [435]. Conversely, HSP70-1A expression in dark-skin melanocytes was found to be higher compared to that with light-skin phenotypes, contributing to skin colour diversity [436]. Ligands of the purinergic receptor type 2 X7 (P2X7) [437], odorant receptor 51E2 [438], olfactory receptor OR2A4/7 [439], aryl hydrocarbon receptor (AHR) [440], and type I cannabinoid receptor (CB1) [441] are also expected to contribute to the regulation of melanogenesis; the expression of these receptors by human melanocytes has been proven, and their stimulation increases the production of melanin.

## 5. Diversity of Human Hair Colour

The diversity in human hair colour results mostly from the quantity and ratio of eumelanin and pheomelanin produced. Mostly, eumelanin is the default hair pigment, being predominant in more than 90% of the human population. Eumelanic phenotypes range from black, dark brown, medium brown, light brown, to blond. All these phenotypes contain small, nearly constant, amounts of pheomelanin (0.85–0.99 μg of melanin/mg of hair), with the varying contents of eumelanin being accountable for the visual differences in hair colour. Dark brown hair reportedly contains eumelanin at 66% (14.6 μg/mg) the level of black hair (22.2 μg/mg), medium brown at 47% (10.4 μg/mg), light brown at 40% (8.7 μg/mg), and blond hair at 23% (4.7 μg/mg). Red hair is the only phenotype that contains comparable amounts of eumelanin and pheomelanin: 3.8 μg/mg and 4.7 μg/mg, respectively [14,15,442]. The activity of TYR (which, as discussed before, is greatly affected by the melanosomal pH) and the cysteine content of melanosomes have been proposed to play critical roles in determining the amount and ratio of the two human hair pigments. Those are influenced by several polymorphisms in a wide range of pigmentary genes.

The red hair colour phenotype, characterized by atypical high pheomelanin contents, is caused mostly by polymorphisms in *MC1R*, the master regulator of pigment-type switching. The most penetrant *MC1R* polymorphisms are R142H, R151C, R160W, and D294H [14,443,444,445,446,447,448]. Such polymorphisms cause low levels of MC1R signalling, leading to the downregulation of various pigmentary genes: *TYR*, *TYRP1*, *OCA2*, *SLC24A5*, and *SLC45A2*; *CTNS* is also expected to be downregulated, leaving high levels of cystine (and consequently, cysteine) inside melanosomes. Therefore, the melanosomes are cysteine-rich, acidic, and with low tyrosinase activity, leading to the production of low-to-medium levels of pheomelanin and eumelanin [14,15,449,450]. Polymorphisms in *ASIP* also show a strong association with red hair [14,15,443,444,451]. *ASIP* encodes the agouti signalling protein, an antagonist of MC1R, which competes with α-MSH for binding to the receptor [289,293,294,452,453,454,455]. By impacting on the acidification of melanosomes, polymorphisms in *SLC45A2*, *SLC24A5*, and *OCA2* may additionally contribute to the red hair phenotype [14,15]. Furthermore, polymorphisms in the cystine/glutamate exchanger *SLC7A11* have also been proposed to be important for pheomelanin production [443].

The blond phenotype is characterized by high levels of MC1R signalling, reduced activities of ion transporters, and full activity of CTNS. These suppressed, but pro-eumelanogenic conditions, make melanosomes acidic and cysteine deficient, leading to the production of trace amounts of pheomelanin and low levels of eumelanin [15]. No single gene has been reported as the main one responsible for this phenotype, with polymorphisms in a number of genes, such as *ASIP*, *KITLG* (encoding for SCF), *MC1R*, *TPCN2* (encoding for TPC2), and *TYRP1*, being associated with blond hair and found to differentiate it from brown hair [14,443,451,456,457,458,459,460,461,462,463,464,465].

The dark hair phenotypes (brown to black) are characterized by high levels of MC1R signalling and full activities of ion transporters and CTNS. Under these conditions, melanosomes become neutral and cysteine deficient, leading to the production of trace amounts of pheomelanin and high levels of eumelanin [15]. Several studies have confirmed an association between dark hair and polymorphisms in genes, such as *SLC24A5*, *SLC45A2*, HECT and RLD domain containing E3 ubiquitin protein ligase 2 (*HERC2*)*,* and interferon regulatory factor 4 (*IRF4*) [14,443,463,466,467].

## 6. Age-Induced Hair Greying

The greying of hair (canities) is one of the most obvious and common signs of aging, occurring to a varying degree in all individuals, regardless of gender or race. A rule of thumb for hair greying is that by 50 years of age, 50% of people have 50% grey hair [51,468]. Besides aging, greying can be a manifestation of a genetic disorder, like the Waardenburg and Griscelli syndromes, where mutations affecting genes involved in the melanogenesis and melanin transfer pathways and their regulation will lead to hypopigmented hair. The term grey hair widely refers to an admixture of pigmented and non-pigmented (white) hairs, although sometimes a single hair fibre can show a progressive dilution from black, through grey to white, over several hair cycles or within the anagen phase of a single cycle [16]. Hair greying usually appears first at the temples, spreading to the vertex and then to the remainder of the scalp, affecting the occiput last [51,468]. At its simplest, the pigment loss in greying hair follicles is due to a marked reduction in melanogenically active melanocytes in the hair bulb. Although it has not yet been completely understood why melanocytes are lost with aging, some mechanisms have been proposed (Figure 9) [468,469,470].

The traditional view proposes that the depletion of hair follicle bulbar melanocytes correlates with oxidative stress. The melanogenic activity of bulbar melanocytes is likely to generate large amounts of reactive oxygen species (ROS), via the oxidation of tyrosine and DOPA to melanin [471]. Although melanocytes possess an efficient antioxidant system, it appears that this defence becomes impaired with age, causing the accumulation of ROS that may generate significant oxidative stress in both melanocytes and the anagen hair bulb epithelium. In this context, melanogenic bulbar melanocytes are best suited to assume a postmitotic, terminally differentiated, (pre)senescence status to prevent cell malignant transformation [469,472,473,474,475]. The involvement of ROS in the onset of canities has been supported by several observations. Some melanosomes from grey hair bulbs were identified within auto-phagolysosomes, suggesting that they are defective and perhaps leaking reactive metabolites. Melanosomes in greying and white hair bulbs are highly vacuolated, a common cellular response to increased oxidative stress. Moreover, common mitochondrial DNA deletion, also a marker of oxidative stress, occurs more prominently in grey hair follicles than in pigmented ones [475].

A depletion of the hair follicle MelSC reservoir has additionally been associated with the onset of greying. On average, a scalp hair follicle experiences 7–15 seedings of melanocytes over the grey-free lifespan. It is possible that canities may reflect an exhaustion of the MelSC reservoir seeding potential; in fact, there is some experimental evidence suggesting that this potential is limited [17,51,476]. The defective, age-related, maintenance of MelSCs, probably due to an imbalance among the antiapoptotic protein BCL2, MITF (its transcriptional regulator), and other factors, may also be accountable for the depletion of the MelSC reservoir [17,65,469,475,477,478,479]. Although many studies have shown the absence of MelSCs in white hair follicles, there are reports of hair greying reversibility. In this context, the greying seems to occur due to some defect in MelSC activation or migration to the hair bulb at the beginning of a new hair cycle [16,51,469].

## 7. Modification of the Hair Fibre Colour

A large proportion of reports on melanogenesis modulation has been dedicated to skin pigmentation, in particular, epidermal melanogenesis inhibition. This is most related to the market pressure, since it is estimated that 15% of the population worldwide invests in skin whitening [480]. Therefore, much of the available information on melanogenesis regulation and its intentional modulation comes from epidermal melanocytes, melanoma cells or more or less complex skin cellular models, and from studies on UV-induced inflammatory skin responses. Though valuable, this information may not necessarily be related to follicular melanogenesis regulation or be useful based on its intentional modulation. In fact, hair colour variation is only partially correlated with skin and eye colour variation, reflecting differences in cellular interactions and melanogenesis regulation in those different tissues [444]. For example, in the skin context, pertinent clinical interest exists to address hyper- and hypopigmentation disorders, many of which have an inflammatory cause; consequently, the development of drugs or treatment methods from the perspective of regulating imbalanced inflammation is appealing [481]. Concerning the modulation of follicular melanogenesis, such strategies are less feasible, because first, anagen terminal hair follicles are less exposed to direct, pro-inflammatory stimulation by UV radiation (as referred in Section 4) and second, healthy anagen follicles possess relative immune privilege from the bulge region downwards to the hair bulb [482]. This means that hair follicles have restricted immune cell recruitment/trafficking ability, reduced antigen presentation, and active immunosuppression [482]. If, on one side, the pathways and mechanisms of epidermal melanogenesis regulation in response to UV may not exhibit the same expression in hair follicles, on the other side, choosing targets and developing strategies to modulate follicular melanogenesis should not involve the modulation of immune effectors, which could interfere with this important immune privilege.

The obvious starting point for the search of potential targets for the intentional modulation of follicular melanogenesis and consequent repercussions on hair fibre colour would be the common synthesis pathway and its enzymes (with the exception of DCT, Section 3.2). Many researchers are devoted to finding direct tyrosinase inhibitors for the management of skin hyperpigmentation. Tyrosinase acts as a rate-limiting enzyme in melanin synthesis (Section 3.2) and it is expressed only by melanocytes; thus, targeting tyrosinase activity would be efficient and without side effects [480]. Many tyrosinase inhibitors were indeed found using mushroom tyrosinase as a model. However, due to the inherent differences between both enzymes, it is not uncommon that the ability of a given molecule to inhibit human tyrosinase falls far short of that seen for mushroom tyrosinase [480]. Determining the 3D structure of human tyrosinase is therefore urgent and will certainly allow for a more focused and efficient search, including in silico studies, for tyrosinase allosteric or orthosteric modulators. Moreover, the direct tyrosinase inhibitors would have to be able to cross both cytoplasmatic and melanosomal membranes in order to act. This bioavailability issue (along with the stability issue in cosmetic formulations) is many times the cause for the disappointing in vivo performances of tyrosinase inhibitors. Bioavailability to the follicular pigmentary units is expected to be higher compared to that to epidermal ones, since the pilosebaceous unit is an extremely relevant skin permeation pathway, especially for nanoparticles; the hair follicles are an entry point for topically applied substances and a privileged target for topical nanodelivery, also functioning as a reservoir due to their anatomical architecture [483]. Finding adequate nanodelivery systems could improve the bioavailability of known tyrosinase inhibitors to follicular melanocytes, which would lead to fibre colour lightening. Another consequence of developing follicular delivery systems targeting anagen hair bulbs will be to widen the range of potential targets and ways to modulate their activities by improving delivery specificity and general safety, restricting side effects.

Other potential targets can be found among the numerous effectors in the signalling pathways regulating melanogenesis [339,484]. If intentional modulation of hair fibre colour is the aim, starting with polymorphic genes, of which single nucleotide polymorphisms have already been strongly associated with particular hair colour phenotypes (Section 5), as potential targets would be a logical approach. Several genome-wide association studies have identified many loci affecting hair colour [444,485]. It has become clear that hair colour is a polygenic phenotype with epistasis phenomena [486]. The intentional modulation of melanogenesis to achieve a given fibre colour may therefore require controlling the activity of more than one target simultaneously. Looking at genes identified with candidate causal variants for blonde, red, and dark brown/black hair colours (Section 5), bioactive molecules capable of directing signalling pathways mediated by MC1R, EDNRB, and KIT receptors or the activity of their ligands, melanosomal membrane transport and pH or melanosome exocytosis are likely to be potential modulators of follicular melanogenesis.

Changes in hair colour have been reported as a side-effect of many drugs used in the treatment of several diseases (Table 2). Those changes occur as either hair lightening or darkening, and even the repigmentation of grey/white hair has been reported. Since such alterations are mostly transient, with the hair returning to its original colour after drug withdrawal, these findings sustain the attractive possibility of using known, safe, and already approved drugs in hair colour modification as an alternative or in addition to the conventional methods. Many of the case reports on hair colour changes as a therapeutic drug side effect have not been further explored in terms of the mechanisms leading to such colour changes. It would be interesting to find the targets of the drugs in hair follicles that are able to affect follicular melanogenesis. Finding a common pattern among therapeutic classes of drugs, known to affect hair pigmentation, would accelerate the discovery of new or improved bioactive molecules.

This drug-based cosmetic approach to natural hair colour modification from the follicle is getting closer and closer to reality as more and more specific and efficient follicular delivery methods emerge and as the intricate process of hair colour formation becomes increasingly unravelled.

## 8. Conclusions

(1)Hair colour is one of the most distinctive human phenotypes, being the subject of extensive studies over the last decades. Hair pigmentation is a manifestation of the presence of pigments called melanins.(2)The colouration of hair shafts results from precise and sequential interactions between different cell populations in the hair follicle, the mini-organ responsible for hair shaft production. Hair pigmentation is strictly coupled to the hair growth cycle, occurring during anagen.(3)Three steps are vital for the uniform and accurate pigmentation of hair shafts: melanosome biogenesis in neural crest-derived melanocytes, the biochemical synthesis of melanins (melanogenesis) in melanosomes, and transfer of melanin granules to surrounding keratinocytes for incorporation into the forming hair fibres. Those steps are under complex genetic control, with MITF being the transcription regulator of many genes associated with those processes.(4)Melanosome biogenesis depends on PMEL, MLANA, and GRP143, which form a protein complex in early stages of maturation assuring the proper organelle composition and structure. The maturation of melanosomes also requires the trafficking of several melanogenic proteins, mediated by adaptor-related protein complexes (AP-1 and AP-3), the biogenesis of lysosomal organelle complexes (BLOC-1 and BLOC-2), SNAREs and RABs.(5)Melanogenesis in vivo produces two chemically distinct types of melanins: the black-to-brown eumelanin and the reddish-brown to yellow pheomelanin. The initiation of melanogenesis requires L-Tyr, which can be directly transported from the extracellular space or synthesized inside melanocytes through the hydroxylation of L-phenylalanine by PAH.(6)Upon uptake into melanosomes (possibly by SLC7A5), L-Tyr is converted to L-DOPA, which is subsequently oxidized to DQ. Both reactions are catalysed by tyrosinase, the most important enzyme in the melanogenic pathway; TYRP1 appears to ensure its appropriate processing and stabilization. In the presence of cysteine, DQ is used in the production of pheomelanin. When most cysteine is depleted, DQ enters the synthetic pathway of eumelanin. The levels of cysteine inside melanosomes are controlled by cystinosin and presumably SLC7A11.(7)Tyrosinase exhibits optimal enzymatic activity at a neutral pH. Since the pH of melanosomes is greatly influenced by their internal ionic equilibrium, its regulation may be achieved through the concerted action of a cascade of ion channels and transporters: V-ATPases, NHEs, OCA2, SLC45A2, SLC24A5, TPC2, and TRPML3, among others.(8)Melanocytes transfer mature melanosomes via dendritic projections. The motility of melanosomes from the perinuclear to the melanocyte periphery occurs along microtubules. The capture and movement of melanosomes in the network of actin filaments of the dendrites is regulated by the RAB27A–MLPH–MYO5A complex.(9)Currently, several models can explain the transference of melanin from melanocytes to keratinocytes: exocytic-mediated transfer model, cytophagocytic-mediated transfer model, fusion model, and membrane vesicle-mediated transfer model. Although the exact mechanism of transfer is still unclear, PAR2-mediated phagocytosis seems to be a mostly necessary step.(10)Melanogenesis is a tightly regulated biochemical pathway. It is under the control of several autocrine, paracrine, and endocrine regulators/factors. POMC-derived peptides α-MSH and β-MSH act on melanocortin receptors, activating the cAMP/PKA pathway, upregulating melanogenesis. The stimulation of melanogenesis by POMC-derived β-END occurs through a PKC-dependent pathway; the activation of PKC is also involved in the melanogenic effect of ET-1 upon binding to EDNRB. Increased synthesis of melanin via the binding of WNT proteins to FDZ receptors involves GSK3β and β-catenin. The positive and negative modulation of melanin synthesis is attained once SCF binds to KIT, with the further activation of MAPK signalling pathways (p38, ERK, JNK). A myriad of neurotransmitters, bone morphogenetic proteins, oestrogens, cell adhesion molecules, and non-coding RNAs has also been implicated in the regulation of melanogenesis.(11)The diversity of human hair colour arises mostly from the quantity and ratio of eumelanin and pheomelanin produced. Mostly eumelanic phenotypes range from black, dark brown, medium brown, light brown, to blond; all of these phenotypes contain small, nearly constant, amounts of pheomelanin with decreasing contents of eumelanin. Red hair is the only phenotype that contains comparable amounts of eumelanin and pheomelanin.(12)The amount and ratio of the two human hair pigments are determined by the activity of TYR and cysteine content of melanosomes, being influenced by several polymorphisms in a wide range of pigmentary genes (MC1R, ASIP, SLC45A2, SLC24A5, OCA2 SLC7A11, KITL, TPCN2, TYRP1, HERC2, IRF4, among others).(13)Grey/white hair is one of the most common signs of aging. It is related to the loss of pigment in hair shafts by virtue of a marked reduction in melanogenically active melanocytes at the hair bulb. Some mechanisms have been proposed for this age-related depletion of bulbar melanocytes: ROS-induced depletion, exhaustion of the MelSC reservoir, and defective activation or migration of MelSCs.(14)Given the enormous social importance attributed to hair colour, the dyeing of hair is today a common practice. Hair dyeing systems are diverse and can be divided into oxidative and non-oxidative, or according to durability, into temporary, semi-permanent, and permanent. On the downside, they have been identified as the source of various adverse health effects and thus, the development of safer ways for hair colour modification is more than ever a pertinent issue.(15)Hair lightening, darkening, and repigmentation have been reported as a side effect of many drugs used in the treatment of several diseases. These findings raise the possibility of interfering with the physiological process of hair pigmentation as an alternative or in addition to conventional methods for hair colour modification. In this sense, the identification and comprehension of the events underlying the pigmentation of hair is an important starting point for the development of such innovative hair cosmetics to change colour from inside out.

## Figures and Tables

**Figure 1 biology-12-00290-f001:**
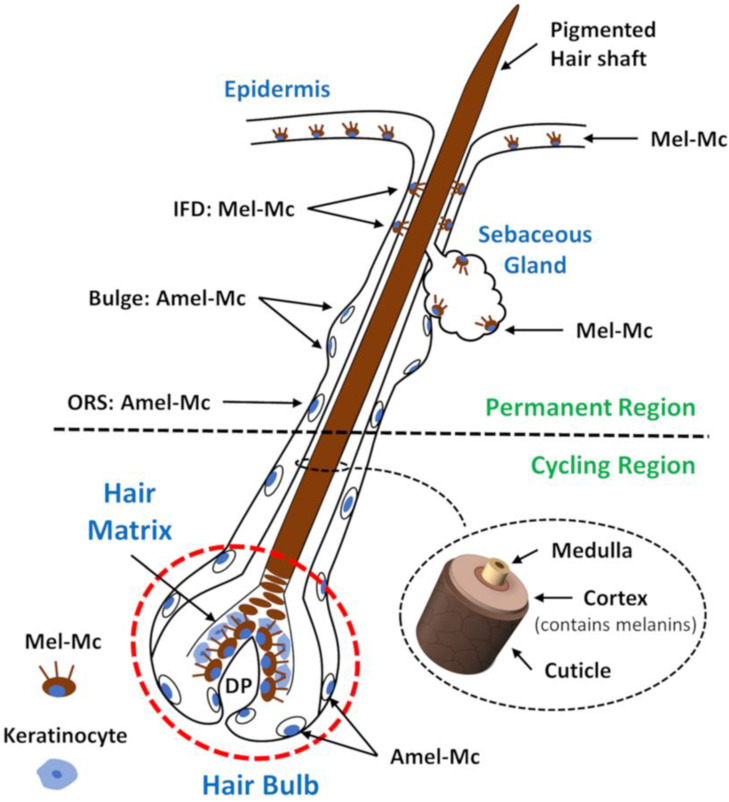
Distribution of melanocytes in the different regions of a human anagen scalp hair follicle. The hair follicle consists of an upper permanent region and an inferior region (containing the hair bulb) that undergoes dramatic cyclic transformation, renewing itself on each hair cycle during our lifetime. Highly melanogenic melanocytes are located above and around the dermal papilla, establishing a functional pigmentary unit with neighbouring pre-cortical keratinocytes that receive melanin granules and incorporate them into the forming fibre cortex. Poorly differentiated melanocytes (amelanotic) are located in the outer-root sheath and bulge regions; they function as a pool of melanocyte stem cells that are recruited on each hair cycle for the regeneration of the pigmentary unit. Amel-Mc: amelanotic melanocytes; DP: dermal papilla; IFD: infundibulum; Mel-Mc: melanogenic melanocyte; ORS: outer root sheath.

**Figure 2 biology-12-00290-f002:**
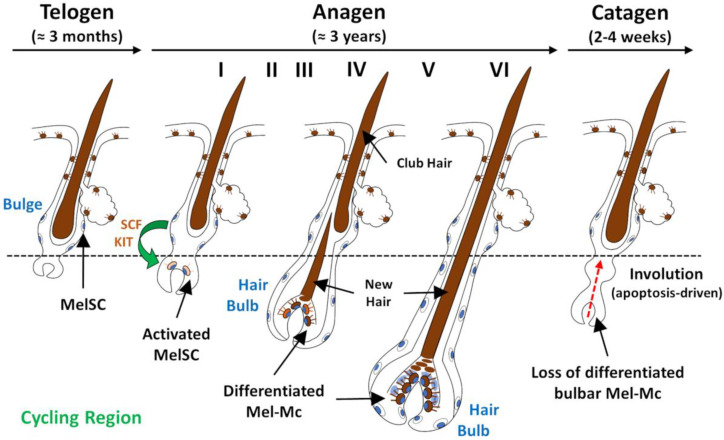
Schematic representation of the fate of melanocytes during the anagen, catagen, and telogen phases of the hair growth cycle. Hair shaft synthesis and pigmentation occur during anagen (or growth phase). Anagen is divided into stages I to VI, throughout which melanocytes experience several morphological and biochemical changes. At the start of anagen, immature melanocytes are stimulated and respond by increasing their volume, entering mitotic activity, and migrating in a coordinate manner, to repopulate the bulb of the new forming hair follicle. By anagen II, melanocytes are in a state of proliferation, which becomes quite significant at anagen III and remains until anagen VI. During the progression of anagen III into VI, melanocytes increase the size and number of their melanosomes and become highly dendritic. At anagen VI, as proliferation ceases, bulbar melanocytes assume a highly differentiated status and become fully functional with respect to melanin synthesis. The catagen phase proceeds through the extensive apoptosis-driven involution of the hair follicle cycling region, followed by telogen, a phase of relative proliferative quiescence. Telogen is characterized by intricate and functionally relevant biochemical activity, focused on the maintenance of several stem cell populations. The active shedding of an old hair fibre (club hair) is a process that occurs as a distinct phase called exogen. KIT: KIT proto-oncogene receptor tyrosine kinase; Mel-Mc: melanogenic melanocytes; MelSC: melanocyte stem cells; SCF: stem cell factor.

**Figure 3 biology-12-00290-f003:**
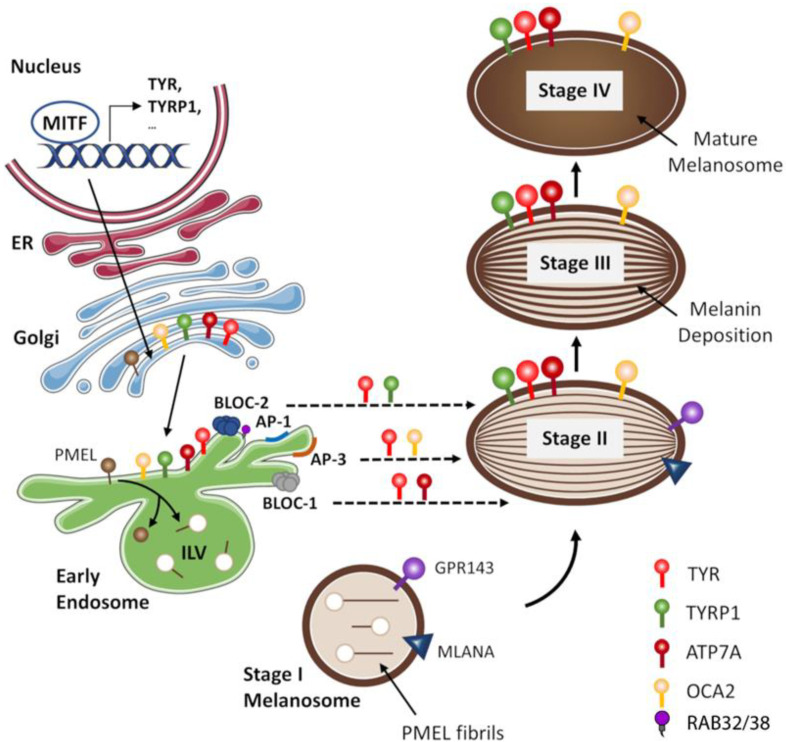
Schematic representation of the biogenesis of an eumelanosome. The development and maturation of melanosomes proceeds through four stages, which can be grouped into two unpigmented (stages I and II) and pigmented (stages III and IV) steps. Melanosomes are assembled in the perinuclear region, near the Golgi stacks, receiving the enzymatic and structural proteins required for melanogenesis. In Stage I, immature melanosomes are spherical, vacuolar domains of early endosomes that harbour intralumenal vesicles formed by invagination of the limiting membrane. In Stage II, pre-melanosomes are characterized by a fully formed melanosome matrix, where PMEL fibrils are organized into arrays of parallel sheets, transforming the spherical Stage I melanosomes into elongated, elliptical organelles. The formation of PMEL fibrils segregates the melanosomal from the endosomal pathways, which seems to involve GPR143 and MLANA. In Stage III, melanogenesis starts with the pigment being regularly and uniformly synthesized on the fibrillar matrix, which causes the darkening and thickening of the matrix fibrils. By Stage IV, the melanosomes are saturated with melanin, in this case predominantly eumelanin, and the internal fibrillar structure becomes completely masked by the pigment; the mature melanosomes are ready to be transferred to adjacent keratinocytes. Melanogenic proteins are loaded into vesicular or tubular carriers for delivery into melanosomes through a process involving the fusion of their limiting membranes. The sorting and transport of melanogenic proteins involve multisubunit protein complexes, such as AP-1 and AP-3, as well as BLOC-1 and BLOC-2. AP-1/3: adaptor-related protein complex 1/3; ATP7A: ATPase copper transporting alpha; BLOC-1/2: biogenesis of lysosomal organelles complex 1/2; ER: endoplasmic reticulum; GPR143: G protein-coupled receptor 143; ILV: intralumenal vesicle; MITF: microphthalmia-associated transcription factor; MLANA: protein melan-a; OCA2: oculocutaneous albinism 2 protein; PMEL: premelanosome protein; TYR: tyrosinase; TYRP1: tyrosinase related protein 1.

**Figure 4 biology-12-00290-f004:**
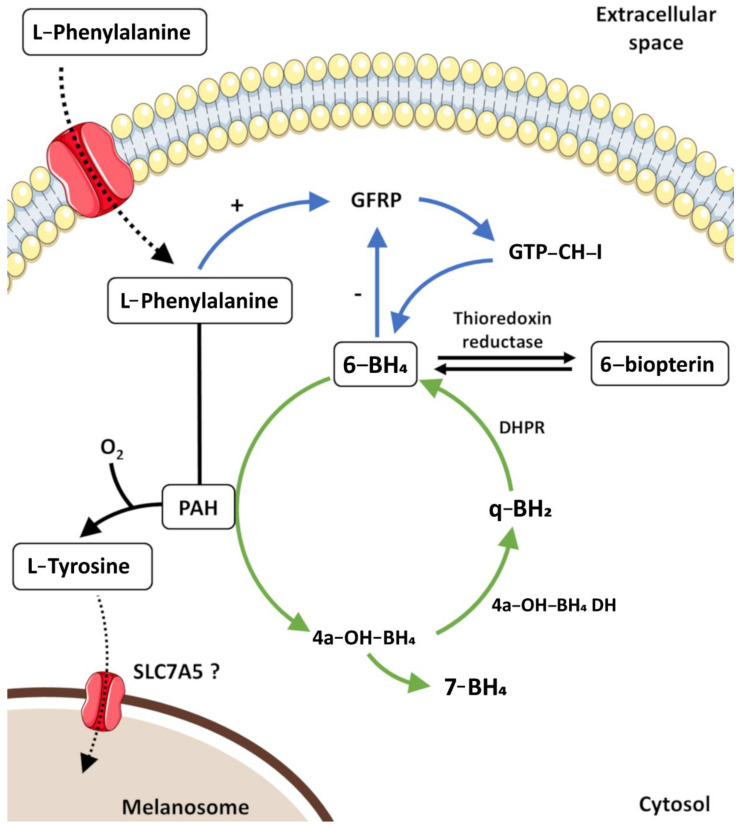
Metabolism of L-phenylalanine to L-tyrosine via phenylalanine hydroxylase (PAH). The initiation of melanogenesis requires L-Tyr, which can be directly transported from the extracellular space or synthesized inside melanocytes through the hydroxylation of L-phenylalanine by PAH. The activity of PAH depends on the co-factor (6R)-L-erythro 5,6,7,8 tetrahydrobiopterin (6-BH4), for which melanocytes have full capacity for de novo synthesis (blue arrows) and recycling (green arrows). Thioredoxin reductase controls the redox status of 6-BH4/6-biopterin, with implications in the regulation of melanin production; reduced 6-BH4 can bind to tyrosinase and inhibit the enzyme, while oxidized 6-biopterin has no effect on it. 4a-OH-BH4 DH: 4a-hydroxy-tetrahydrobiopterin dehydratase; 4a-OH-BH4: 4a-hydroxy-tetrahydrobiopterin; 7-BH4: (7R)-L-erythro-5,6,7,8-tetrahydrobiopterin; DHPR: dihydropteridine reductase; GFRP: GTP-cyclohydrolase I feedback regulatory protein; GTP-CH-I: GTP-cyclohydrolase I; q-BH2: quinonoid dihydropterin; SLC7A5: solute carrier family 24, member 5.

**Figure 5 biology-12-00290-f005:**
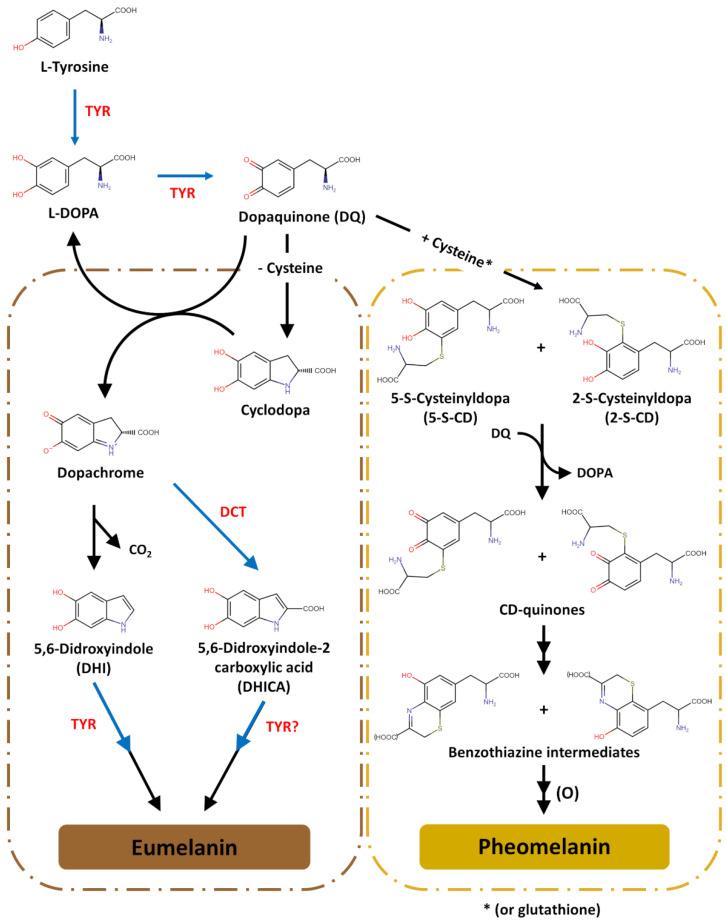
Biosynthetic pathways involved in the production of eumelanin and pheomelanin in human melanosomes. In these organelles, melanogenesis begins with the synthesis L-DOPA from L-Tyr and subsequent oxidation to DQ. The hydroxylation of L-Tyr to L-DOPA is catalysed by the multi-functional enzyme TYR. From here, mixed melanogenesis proceeds, using the same precursor (DQ) for the synthesis of pheomelanin and eumelanin. Pheomelanin is spontaneously produced from DQ if cysteine is present at concentrations above 1 μM. The first step is the reductive addition of cysteine to DQ, giving rise to CD isomers 5-S-CD and 2-S-CD, at a ratio of 5 to 1. The second step is the redox exchange between CD isomers and DQ to produce CD-quinones (and DOPA), followed by their cyclization through dehydration to form ortho-quinonimine (QI). The last step involves the polymerization of benzothiazine intermediates to pheomelanin. The production of pheomelanin is preferred over the production of eumelanin in the presence of CD isomers at concentrations above 10 μM. The production of eumelanin begins after most CD isomers and cysteine are depleted. DQ undergoes spontaneous intramolecular cyclization to give rise to cyclodopa. Then, cyclodopa rapidly undergoes redox exchange with another DQ molecule to produce one molecule of dopachrome, a relatively stable intermediate, and one molecule of DOPA. In the absence of additional factors, dopachrome undergoes spontaneous decarboxylative rearrangements to form 5,6-dihydroxyindole (DHI). In the presence of DCT, the tautomerization of dopachrome can also form 5,6-dihydroxyindole-2-carboxylic acid (DHICA). However, human bulbar melanocytes (at least those of eumelanic phenotypes) do not express this protein, and human hairs contain low yields of DHICA. Both dihydroxyindoles (DHI and DHICA) are further oxidized and assembled via cross-linking reactions into eumelanin polymers. Oxidative polymerization of DHI is catalysed directly by tyrosinase and indirectly by DQ. Spontaneous reactions are denoted as black arrows. Enzyme-catalysed reactions are signalized by blue arrows. CD: cysteinyldopa; L-DOPA: L-3,4-dihydroxyphenylalanine; TYR: tyrosinase; DCT: dopachrome tautomerase.

**Figure 6 biology-12-00290-f006:**
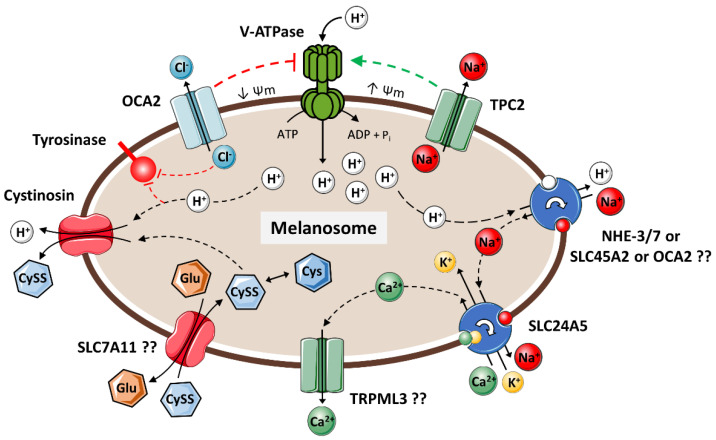
Schematic representation of the cascade of ion channels and transporters with a role in melanogenesis. Melanosome acidification, needed in the early stages of melanosome maturation, is accomplished by V-ATPases. To increase the melanosomal pH, which allows for melanogenesis, the H^+^ protons are returned to the cytoplasm through sodium–proton exchangers, namely NHE-3 and NHE-7. OCA2 is a chloride (Cl^-^) channel that reduces proton import into melanosomes, neutralizing the luminal pH and activating the pH-sensitive TYR enzyme. SLC45A2 has also been proposed to act as proton/glucose exporter, respectively. The excess of Na^+^ needs to be cycled out of the melanosomes, which involves SLC24A5. The exchanger activity of SLC24A5 also provides a link between cytosolic and melanosomal Ca^2+^. TPC2 was found to function as an Na^+^-selective channel, providing the negative regulation of melanogenesis by increasing melanosomal membrane potential and acidity, possibly by providing a cation counterflux to enhance the H^+^ transport by V-ATPases. The counterbalance of Ca^2+^ influx can be accomplished by TRPML3, to the neutralization of melanosomal pH. Cystinosin is a cystine/H^+^ symporter that controls the efflux of cystine; its ability to efflux protons along with cystine also renders it as having a role in the control of melanosomal pH. Moreover, the coordinated activities of both cystinosin and SLC7A11 may determine the levels of melanosomal cysteine, therefore influencing the eumelanin/pheomelanin ratio. Cys: cysteine; CySS: cystine; Glu: glutamate; NHE: sodium hydrogen exchanger; OCA2: oculocutaneous albinism 2 protein; SLC(7A11/24A5/45A2): solute carrier family 7/24/45, member 11/5/2; TPC2: two pore segment channel 2; TRPML3: transient receptor potential cation channel, mucolipin subfamily, member 3; V-ATPase: vacuolar ATPase.

**Figure 7 biology-12-00290-f007:**
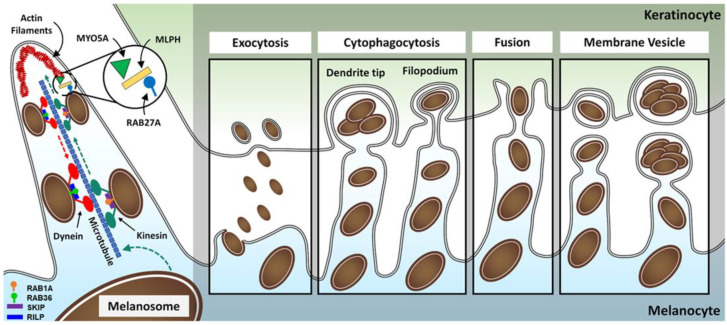
Mechanisms of melanosome transport and transfer. Melanosomes bind kinesin to move along microtubules from the perinuclear region to the periphery of melanocytes. This anterograde movement is regulated by RAB1A and its effector protein SKIP. At the dendrites, the movement of melanosomes in the rich network of actin filaments is regulated by the RAB27A–MLPH–MYO5A complex. Retrograde microtubule-dependent melanosome transport (dynein-dependent) is regulated by RAB36 and its effector protein RILP. Several models have been proposed regarding the transfer of melanin from melanocytes to surrounding keratinocytes; the most recent evidence points towards the exocytosis/endocytosis and membrane vesicle shedding models. MLPH: melanophilin; MYO5A: myosin VA; RAB1A: Ras-related protein RAB1A; RAB27A: Ras-related protein RAB27A; RAB36: Ras-related protein RAB36; RILP: Rab interacting lysosomal protein; SKIP: skeletal muscle and kidney enriched inositol phosphatase.

**Figure 8 biology-12-00290-f008:**
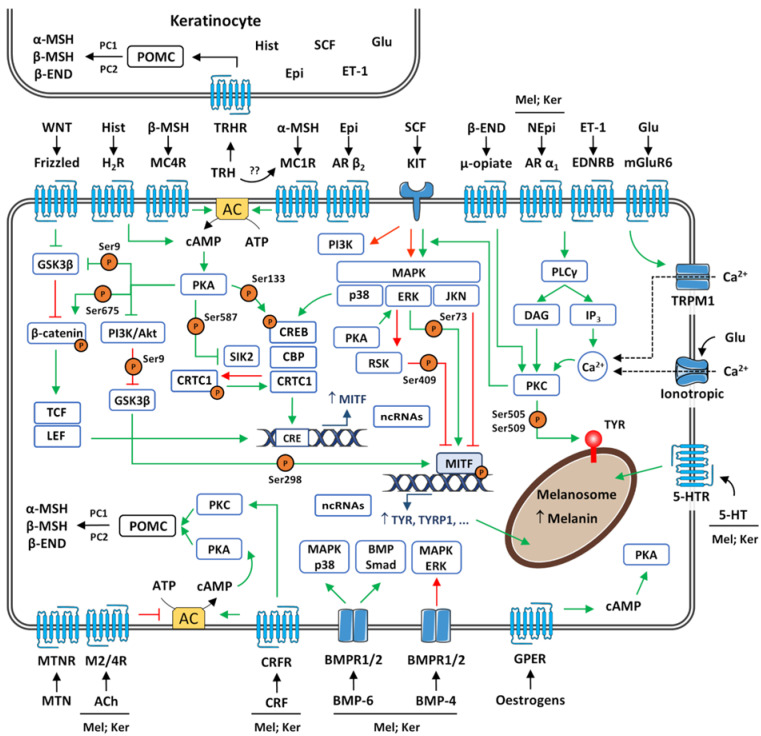
Major molecular pathways involved in the regulation of melanin synthesis in melanocytes. Green lines denote positive regulation of melanogenesis. Red lines refer to negative regulation of melanogenesis. α/β-MSH: α/β-melanocyte-stimulating hormone; β-END: β-endorphin; 5-HT(R): serotonin (receptor); AC: adenylyl cyclase; Ach: acetylcholine; Akt: serine/threonine kinase; AR: adrenergic receptor; BMP-4/6: bone morphogenetic protein 4/6; BMPR1/2: bone morphogenetic protein receptors 1/2; cAMP: cyclic adenosine monophosphate; CBP: CREB-binding protein; CRE: cAMP responsive element; CREB: cAMP response element-binding protein; CRF(R): corticotropin-releasing factor (receptor); CRTC1: CREB-regulated transcription coactivator 1; DAG: diacylglycerol; EDNRB: endothelin receptor type B; ERK: extracellular signal-regulated kinases; ET-1: endothelin 1; GPER: G protein-coupled oestrogen receptor; GSK3β: glycogen synthase kinase 3β; H2R: histamine H2 receptor; Hist: histamine; IP3: inositol–triphosphate; JNK: c-Jun N-terminal kinases; Ker: keratinocytes; KIT: KIT proto-oncogene, receptor tyrosine kinase; LEF: lymphoid enhancer-binding factor; M2/4R: muscarinic 2/4 receptor; MAPK: mitogen-activated protein kinases; MC1/4R: melanocortin 1/4 receptor; Mel: melanocytes; m(Glu)R6: metabotropic (glutamate) receptor 6; MITF: microphthalmia-associated transcription factor; MTN(R): melatonin (receptor); ncRNAs: non-coding ribonucleic acids; (N)Epi: (nor)epinephrine; PC1/2: proprotein convertase 1/2; PI3K: phosphatidylinositol 3-kinase; PKA/C: protein kinase A/C; PLCγ: phospholipase Cγ; POMC: pro-opiomelanocortin-derived peptides; RSK: p90 ribosomal S6 kinase; SCF: stem cell factor (SCF); SIK2: salt-inducible kinase 2; TCF: T-cell factor; TRH(R): thyrotropin-releasing hormone (receptor); TRPM1: transient receptor potential cation channel, subfamily M, member 1; TYR: tyrosinase; TYRP1: tyrosinase related protein 1.

**Figure 9 biology-12-00290-f009:**
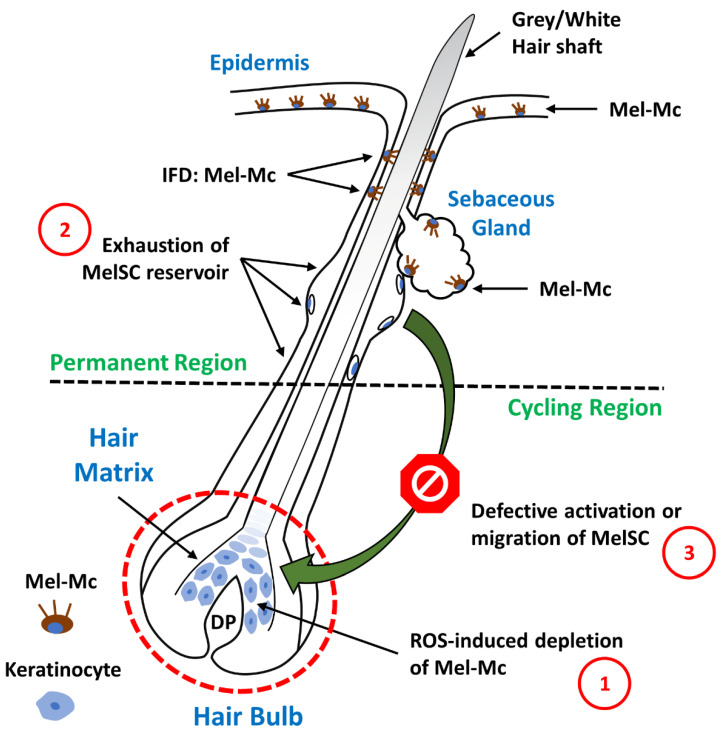
Proposed causes for hair greying during aging. The age-related pigment loss in hair fibres may be caused by the following: (1) depletion of melanotic melanocytes (Mel-Mc) in the hair matrix. Intrinsic (melanogenesis generates ROS) and extrinsic (inner root sheath and pre-cortical keratinocyte terminal differentiation involves ROS generation) sources of ROS may lead to oxidative damage of bulb melanocytes if antioxidant systems start to fail. (2) Exhaustion of the melanocyte stem cell (MelSC) reservoir, and (3) MelSC defective activation or migration during the onset of the anagen phase of the hair growth cycle. The inability to maintain MelSC quiescence due to ageing, oxidative stress, and stress-induced sympathetic nervous activity results in ectopic MelSC differentiation. Differentiated melanocytes are no longer able to replenish the pigmentary unit in the beginning of the hair cycle.

**Table 1 biology-12-00290-t001:** Most representative genes involved in the pigmentary process, with expression controlled by the transcription factor MITF. DHI: 5,6-dihydroxyindole; L-DOPA: L-3,4-dihydroxyphenylalanine.

Symbol	Gene Name	Protein Function	Reference(s)
Melanosomes Biogenesis			
PMEL	Premelanosome protein	Amyloid fibril essential for melanosome biogenesis	[69]
MLANA	Melan-a	Regulation of PMEL expression	[69]
GPR143	G protein-coupled receptor 143	Regulation of melanosome maturation	[70]
Biochemical Synthesis of Melanin			
TYR	Tyrosinase	Hydroxylation of L-tyrosine to L-DOPAOxidative polymerization of DHI	[71,72,73,74]
TYRP1	Tyrosinase related protein 1	Stabilization of tyrosinase	[73,75]
DCT	Dopachrome tautomerase	Tautomerization of dopachrome	[75]
Regulation of Melanosomal pH			
SLC24A5	Solute carrier family 24 member 5	Na^+^/H^+^ exchanger	[76]
SLC45A2	Solute carrier family 45 member 2	K^+^-dependent Na^+^/Ca^2+^ exchanger	[77]
Melanosome Transport			
RAB27A	RAB27A, member RAS oncogene family	Small GTPase able to bind to a variety of effector proteins; regulates melanosome trafficking in melanocytes	[78]
Regulation of Melanogenesis			
MC1R	Melanocortin 1 receptor	Mediates melanocortin signalling	[79]
KIT	KIT proto-oncogene, receptor tyrosine kinase	Mediates stem cell factor signalling	[76]
EDNRB	Endothelin receptor type B	Mediates endothelin-1 signalling	[80]

**Table 2 biology-12-00290-t002:** Drugs reported to change the colour of hair as a side effect of medical treatments.

Drug	Reason for Use	Hair Change	Reference(s)
α-Interferon	Melanoma	Lightening	[487]
Acitretin	Psoriasis	Repigmentation	[488,489]
Atezolizumab	Lung cancer	Repigmentation	[490]
Chloroquine	Malaria prophylaxis	Lightening	[491]
	Dermatomyositis		[492]
Cisplatin	Metastatic germ cell neoplasm of the testis	Darkening	[493]
		Lightening	
Cyclosporin	Psoriasis	Darkening	[494]
Dabrafenib	Metastatic melanoma	Repigmentation	[495]
Dasatinib	Chronic myeloid leukaemia	Lightening	[496]
Defibrotide	Deep venous thrombosis	Darkening	[497]
Dupilumab	Atopic dermatitis	Repigmentation	[498]
Erlotinib	Lung adenocarcinoma	Repigmentation	[499]
Etretinate	Psoriasis	Lightening	[500]
		Repigmentation	[501]
	Pityriasis rubra pilaris		[502]
Hydroxychloroquine	Discoid lupus erythematosus	Lightening	[503]
Imatinib mesylate	Gastrointestinal stromal tumour	Lightening	[504]
	Chronic myeloid leukaemia		[505]
L-Thyroxine	Hypothyroidism	Repigmentation	[506]
Latanoprost	Open-angle glaucoma	Repigmentation	[507]
Lenalidomide	Multiple myeloma	Repigmentation	[508]
Levodopa	Parkinson’s disease	Repigmentation	[509]
Nivolumab	Lung cancer	Repigmentation	[490]
*Para*-aminobenzoic acid	Lymphoblastoma cutis	Darkening	[510]
	Dermatomyositis		
	Dermatitis herpetiformis		
	Scaly erythroderma		
	Scleroderma		
Pazopanib	Hürthle cell carcinoma	Lightening	[511]
	Myofibroblastic sarcoma		[512]
Pembrolizumab	Lung cancer	Repigmentation	[490]
Prednisone	Bullous pemphigoid	Repigmentation	[513]
Selumetinib	Plexiform neurofibroma	Lightening	[514]
Sunitinib malate	Gastrointestinal stromal tumour	Lightening	[515]
Tamoxifen	Breast cancer	Repigmentation	[516]
Thalidomide	Multiple myeloma	Repigmentation	[517]
Triptorelin	Precocious puberty	Greying	[518]
Ustekinumab	Psoriasis vulgaris	Repigmentation	[519]
Valproic acid	Seizures	Lightening	[520]
Vemurafenib	Metastatic melanoma	Repigmentation	[495]
Verapamil	Hypertension	Repigmentation	[521]

## Data Availability

Not applicable.

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
