# Peer review of "A Comprehensive Review of Mammalian Pigmentation: Paving the Way for Innovative Hair Colour-Changing Cosmetics"

_biology, 2023, doi:10.3390/biology12020290_

Round 1

Reviewer 1 Report

The review by Fernandes et al. is comprehensive and complete, focusing on all the processes involved in hair coloring determination. It is very well written and easy to follow, despite some minor typos. However, the authors need to review the literature more critically, convey their views on the published studies and suggest future research priorities. The Figures are also a very good complement to the text and convey the main messages effectively. Nevertheless, there are a number of important points to be addressed:

1 -  In several paragraphs, only the last sentence has references. If the references of the last sentence refer to all the sentences of the paragraph, they should appear after the respective sentence. Therefore, the whole manuscript must be revised so that all the sentences that need, have references at the end.

2 - In the text, the references go up to 360 and the list at the end of the manuscript only has 297. Therefore, this should be corrected.

3 - Figure legends are very minimalistic, in general, and mainly define the acronyms. Therefore, they should be expanded to be more informative (the message of each figure must be clear from the reading of the legend). 

4 - Not all the acronyms are defined (e.g., LAMP, CREB) and, normally, the acronyms appears in brackets after being defined (and not the other way around).

5 - In section 41.1, it is not referred that melanosomes are lysosome-related organelles and this must be associated to the characteristics described.

6 - The post-Golgi transport of tyrosinase and TYRP1, which is regulated by Rab32 and Rab38 must be shown in Fig. 3 and referred in the text.

7 - The role of the other Rabs referred (7, 9, 11 and 33A) has to be (at least briefly) described. It is not enough to refer that they are "involved in the trafficking of melanogenic proteins, as well as other aspects of melanosome biogenesis". And what are the other Rabs?

8 - In section 4.2 (1st paragraph), the authors should explain how the ratio of pheomelanin to eumelanin is influenced by the availability of cysteine, as well as the structural differences between pheomelanosomes and eumelanosomes (and point out that Fig. 3 and the accompanying text describe the biogenesis of eumelanosomes).

9 - Regarding Fig. 7, it does not have a legend and several aspects need improvement: In the case of the exocytosis model, melanin is surrounded by a single membrane inside keratinocytes; Rab1a-SKIP-Kinesin 1 and Rab36-RILP-dynein could be represented (in this case, the figure should be cited in line 661; citation should also be added in the sentence that ends in line 655).

10 - In section 4.3.1, the authors should try to explain why there is evidence for different transfer models and be more critical, as there is overwhelmingly more evidence to support the exocytosis-mediated transfer model. Also, the authors failed to refer some of the regulators already described for the exocytosis of melanosomes, including Rab11b and Rab3a, as well as for some of the other models (e.g., Rab17 and NMDA receptor for filopodia-mediated transfer and RhoA for the membrane vesicle-mediated transfer).

11 - The authors refer the transfer through filopodia in the cytophagocytosis- and membrane fusion-mediated transfer models. Indeed, the tips of filopodia could be phagocytosed or fuse with the membrane of keratinocytes to allow the transfer of melanosomes directly through these structures. Therefore, this should be explained and the possibility that these two models are variations of the same, referred.

12 - In section 7 (and Fig. 9), it must be considered the possibility that grey hair could be caused by defects in melanin transfer. Indeed, Griscelli syndrome patients and mouse models show silvery hair/fur and the defect is thought to be on melanin transfer. 

13 - Some of the information that is given in section 8 could be part of the Introduction since the authors state why they wrote this review and why it is important to better understand the molecular mechanism of hair pigmentation. 

14 - When the authors refer that the long-term usage of hair dyes can cause side effects and increase the risk of developing certain cancers, they should carefully explain this statement (what types of cancer, what are the side effects and what is the evidence), as this is a strong statement and can leave the readers wondering about the exact dangers of hair dying.

15 - Examples of targets for the modification of hair color (line 1188) should be given. Indeed, the authors do not go far enough in suggesting future research priorities and suggestions regarding this statement.

Minor points:

1 - What are the changes in the local milieu the author refer in line 167? These should be specified.

2 - In table 1, "melanosome trafficking" is not very specific and "in brain" is missing after "Ras-related protein". Also, reference 10 is from 2008 and there are several references published years before this one about the subject.

3 - In line 675, "transference" should be replaced by "transfer".

4 - In line 713, the direction (from where to where) of melanosome transport should be specified.

5 - In line 751, it would be better to refer why the position of follicular melanocytes is more protected.

6 - In the legend of Fig. 8, the meaning of the colors of the boxes needs to be explained. 

7 - In line 837, the authors could refer where is POMC expressed.

8 - In line 891, can the authors refer in which section was it discussed?

9 - TRP1 and TYRP1 appear written differently. The authors should be consistent throughout the manuscript.

10 - In section 5.9, "mRNA" could be omitted when it appears between brackets) to make it easier to read (and shorter).

11 - Finally, the manuscript should be reviewed for several typos. 

Reviewer 2 Report

This review article describes this interesting biological process of hair coloring and the possibility to tune hair color by interfering with the biology. The review is very comprehensive, perhaps a little too comprehensive that it started to loose focus. The overall quality of the review article is great but it could use some tailoring to become more focused on the topic. I think a review article with less than 30 pages would be sufficient to cover this topic without losing the reader. My detailed suggestions are listed below:

1. The rationale for the review, based on the abstract, is that current hair dyes can cause serious health issues and it's possible to change the hair color by interfering with its biology. The introduction part should expand on the rationale and set up the importance of the review. The discussion on the evolutionary pressures for hair pigments is less relevant for the topic.

2. The review does not need to be a textbook on skin biology, but should be specifically focused on which proteins/biological process can be targeted to change hair color. As a result, section 2 (how melanoblasts become melanocytes during embryogenesis) and section 3 (the hair growth cycle) are less relevant, and should be truncated if not totally removed.

3. Like many other biological process, melanosome biogenesis (4.1) and melanogenesis (4.2) involve many proteins to perform their functions. This review should be focused on those protein targets that can be modulated. For example, tyrosinase can be targeted and there has been extensive efforts to develop tyrosinase inhibitors for skin whitening. In contrast, I don't see any way to modulate melanosomal pH even though it affects pigmentation.

4. The same is true with section 5. MC1R is an obvious target for hair pigmentation, whereas people wouldn't try to modulate the Wnt signaling or adrenergic receptors for hair color change because these signaling pathways are involved in many other biological processes. This section needs to be more focused.

5. Section 4.3 if the exact mechanism of melanin transfer is not clear, the readers don't need to know the details of 4 different models because it is too specific and not as relevant to the topic.

6. As a take home message, the readers would appreciate a table with clinically relevant biological targets, whether or not there has already been drugs targeting this target and what color change can be achieved by modulating the target. 

Round 2

Reviewer 1 Report

The authors did a good job in addressing the reviewers' criticisms. As a result, the review is now more complete and accurate. However, a few (minor) points remain to be addressed:

1 - No reference to lysosome-related organelles could be found in the revised version of the manuscript.

2 - It is essential to feature the post-Golgi transport of tyrosinase and TYRP1 to melanosomes, regulated by Rab32 and Rab38, as this is an alternative pathway of delivery of melanogenic enzymes.

3 - The authors removed the reference to Rabs 7, 9, 11 and 33A instead of briefly describing their role in melanosome biogenesis. It is not necessary to dedicate one paragraph or even several lines to each one (only a few words regarding their functions will suffice).

4 - In the sentence that starts in line 1268, melanosome transfer should be added to the processes that cause hair greying, as this is what occurs in GS.

5 - In the legend of Fig. 8, instead of stating that the colors of the boxes were randomly chosen, authors could simply omit the colors.

Author Response

Dear reviewer 1,

We have replied to your comments as followed:

1 - No reference to lysosome-related organelles could be found in the revised version of the manuscript.

We apologise, probably, the modification was not saved and somehow it was lost without being noticed by us. The expression “lysosome-related organelles” was introduced in the 1st paragraph of section now numbered 3.1.

2 - It is essential to feature the post-Golgi transport of tyrosinase and TYRP1 to melanosomes, regulated by Rab32 and Rab38, as this is an alternative pathway of delivery of melanogenic enzymes.

The addition was made as suggested.

3 - The authors removed the reference to Rabs 7, 9, 11 and 33A instead of briefly describing their role in melanosome biogenesis. It is not necessary to dedicate one paragraph or even several lines to each one (only a few words regarding their functions will suffice).

The requested information was added except for Rab33a, which was cited in the first version, but it does not have yet a known and clear function in pigmentation.

4 - In the sentence that starts in line 1268, melanosome transfer should be added to the processes that cause hair greying, as this is what occurs in GS.

The correction was made as suggested.

5 - In the legend of Fig. 8, instead of stating that the colors of the boxes were randomly chosen, authors could simply omit the colors.

The colours were omitted as suggested.

Reviewer 2 Report

I see no improvement in the revised manuscript. It turns from 44 pages into 68 pages. Personally, I enjoy review papers that are focused and succinct so that I can gather relevant information without being distracted. However, it depends on the style and the policy of a journal. If the journal policy allows a review paper of 68 pages to be published and the editor is ok with it, I am ok with publishing this paper.

Author Response

Dear reviewer 2,

The body length of the manuscript is the same (35 pages), the difference is that the reference list was incomplete in the first version and now it was corrected. We regret that you feel that there was no improvement; we respect your opinion, but more importantly, for us, is the content of the manuscript which was further improved.